# Kernel Language Entropy: Fine-grained Uncertainty Quantification for LLMs from Semantic Similarities

**Alexander Nikitin**[1]    **Jannik Kossen**[2]    **Yarin Gal**[2]    **Pekka Marttinen**[1]

[1] Department of Computer Science, Aalto University
[2] OATML, Department of Computer Science, University of Oxford
`alexander.nikitin@aalto.fi`

## Abstract

Uncertainty quantification in Large Language Models (LLMs) is crucial for applications where safety and reliability are important. In particular, uncertainty can be used to improve the trustworthiness of LLMs by detecting factually incorrect model responses, commonly called hallucinations. Critically, one should seek to capture the model's *semantic uncertainty*, i.e., the uncertainty over the *meanings* of LLM outputs, rather than uncertainty over lexical or syntactic variations that do not affect answer correctness. To address this problem, we propose *Kernel Language Entropy* (KLE), a novel method for uncertainty estimation in white- and black-box LLMs. KLE defines positive semidefinite unit trace kernels to encode the *semantic similarities* of LLM outputs and quantifies uncertainty using the von Neumann entropy. It considers pairwise semantic dependencies between answers (or semantic clusters), providing more fine-grained uncertainty estimates than previous methods based on hard clustering of answers. We theoretically prove that KLE generalizes the previous state-of-the-art method called semantic entropy and empirically demonstrate that it improves uncertainty quantification performance across multiple natural language generation datasets and LLM architectures.

## 1 Introduction

Large Language Models (LLMs) have demonstrated exceptional capabilities across a wide array of natural language processing tasks [58, 66, 69]. This has led to their application in many domains, including medicine [11], education [32], and software development [40]. Unfortunately, LLM generations suffer from so-called hallucinations, commonly defined as responses that are "nonsensical or unfaithful to the provided source content" [26, 18, 52]. Hallucinations pose significant risks when LLMs are deployed to high-stakes applications, and methods that reliably detect them are sorely needed.

A promising direction to improve the reliability of LLMs is *estimating the uncertainty* of model generations [36, 13, 51, 44, 23]. For instance, high predictive uncertainty is indicative of model errors or hallucinations in settings such as answering multiple-choice questions [30]. This allows us to prevent harmful outcomes by abstaining from prediction or by consulting human experts. However, the best means of estimating uncertainty for free-form natural language generation remains an active research question. The unique properties of LLMs and natural language preclude the application of established methods for uncertainty quantification [20, 39, 45, 59, 55].

A particular challenge is that language outputs can contain multiple types of uncertainty, including lexical (which word is used), syntactic (how the words are ordered), and semantic (what a text means). For many problems, *semantic* uncertainty is the desired quantity, as it pertains directly to the accuracy of the meaning of the generated response. However, measuring the uncertainty of the generation via token likelihoods conflates all types of uncertainty. To address this, Kuhn et al. [36] have recently introduced semantic entropy (SE), which estimates uncertainty as the predictive entropy of generated texts with respect to clusters of identical semantic meaning (we discuss this in more detail in Sec. 2).

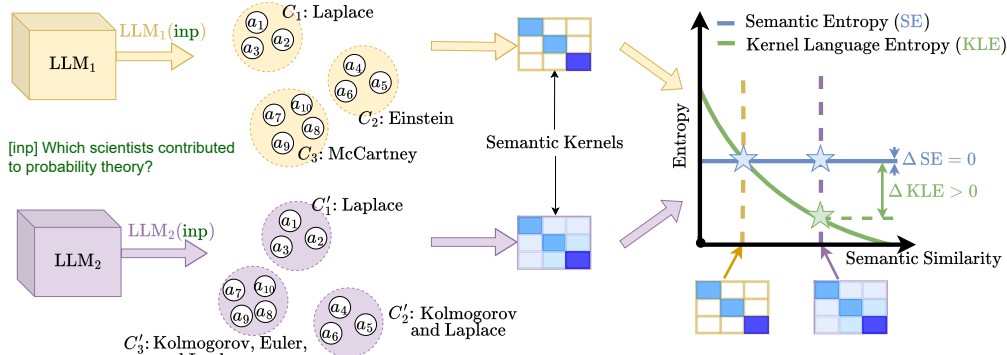

Figure 1: Illustration of Kernel Language Entropy (KLE). We here show a version of KLE called KLE-c, which operates on semantic clusters. Given an input query and two different LLMs, we sample 10 answers from each model $a_1, \ldots, a_{10}$ and $a'_1, \ldots, a'_{10}$ and cluster them by semantic equivalence into clusters $C_1, \ldots, C_3$ and $C'_1, \ldots, C'_3$. For the sake of the example, we assume that the numbers and sizes of clusters, as well as individual cluster probabilities, are all equal $p(C_i|\text{inp}) = p(C'_i|\text{inp})$ for all $i$. Then, semantic entropy would yield identical uncertainties for both LLMs. However, uncertainty should be lower for $\text{LLM}_2$ because semantic "similarity" between the generations is much higher; i.e., the model is fairly confident that "Kolmogorov" and "Laplace" are good answers. KLE, explicitly accounts for the semantic similarity between texts using a kernel-based approach. Semantic kernels provide an effective way to encode the semantic similarity between answers, enabling the method to correctly identify that $\text{LLM}_2$'s outputs should be assigned lower uncertainty (see right).

A critical limitation of SE is that it captures semantic relations between the generated texts only through equivalence relations. This does not capture a *distance metric* in the semantic space, which would allow one to account for more nuanced *semantic similarity* between generations. For instance, it separates "apple" as equally strongly from "house" as it will "apple" from "granny smith" even though the latter pair is more closely related. In this paper, we address this problem by incorporating a distance in the semantic space of generated answers into the uncertainty estimation.

We propose **Kernel Language Entropy (KLE)**. KLE leverages semantic similarities by using a distance measure in the space of the generated answers, encoded by unit trace positive semidefinite kernels. We quantify uncertainty by measuring the von Neumann entropy of these kernels. This approach allows us to incorporate a metric between generated answers or, alternatively, semantic clusters into the uncertainty estimation. Our approach uses kernels to describe semantic spaces, making KLE more general and better at capturing the semantics of generated texts than the previous methods. We theoretically prove that our method is more expressive than semantic entropy, meaning there are cases where KLE, but not SE, can distinguish the uncertainty of generations. Importantly, our approach does not rely on token likelihood and works for both white-box and black-box LLMs.

Our work makes the following contributions towards better uncertainty quantification in LLMs:

- We propose Kernel Language Entropy, a novel method for uncertainty quantification in natural language generation (Sec. 3),
- We propose concrete design choices for our method that are effective in practice, for instance, graph kernels and weight functions (Sec. 3.2),
- We prove that our method is a generalization of semantic entropy (Thm. 3.5),
- We empirically compare our approach against baselines methods across several tasks and LLMs with up to 70B parameters (60 scenarios total), achieving SoTA results (Sec. 5).

We release the code and instructions for reproducing our results at https://github.com/AlexanderVNikitin/kernel-language-entropy.

## 2 Background

**Uncertainty Estimation.** Information theory [49] offers a principled framework for quantifying the uncertainty of predictions as the predictive entropy of the output distribution:

$$\text{PE}(x) = H(Y \mid x) = -\int p(y \mid x) \log p(y \mid x) dy, \tag{1}$$

where $Y$ is the output random variable, $x$ is the input, and $H(Y|x)$ is a conditional entropy which represents average uncertainty about $Y$ when $x$ is given. Uncertainty is often categorized into aleatoric (data) and epistemic (knowledge) uncertainty. Following previous work on uncertainty quantification in LLMs, we assume that LLMs capture both types of uncertainty [30] and do not attempt to disambiguate them, as both epistemic and aleatoric uncertainty contribute to model errors.

**UQ in sequential models.** Let $S \in \mathcal{T}^N$ be a sequence of length $N$, consisting of tokens, $s_i \in \mathcal{T}$, where the set $\mathcal{T}$ denotes a vocabulary of tokens. The probability of $S$ is then the joint probability of the tokens, obtained as the product of conditional token probabilities:

$$p(S \mid x) = \prod_i p(s_i \mid s_{<i}, x). \tag{2}$$

Instead of Eq. (2), the geometric mean of token probabilities has proven to be successful in practice [50]. Using Eq. (1) and (2), we can define the predictive entropy of a sequential model.

**Definition 2.1.** *The predictive entropy for a random output sequence $S$ and input $x$ is*

$$U(x) = H(S \mid x) = -\sum_s p(s \mid x) \log(p(s \mid x)), \tag{3}$$

*where the sum is taken over all possible output sequences $s$.*

A downside of naive predictive entropy for Natural Language Generation (NLG) is that it measures uncertainty in the space of tokens while the uncertainty of interest lies in semantic space. As an illustrative example, consider two sets of $n$ answers, $S_i$ and $S_i'$ sampled from two LLMs with equivalent token likelihood $p(S_i|x) = p(S_i'|x)$ as a response to the question "What is the capital of France?" [36]. Suppose the answers from the first LLM are various random cities ("Paris", "Rome", etc.), and those from the second LLM are paraphrases of the correct answer "It is Paris". Naive predictive entropy computation can give similar values, even though the second LLM is not uncertain about the meaning of its answer. Kuhn et al. [36] have proposed semantic entropy to address this problem.

We first define the concept of semantic clustering. Semantic clusters are equivalence classes obtained using a semantic equivalence relation, $E(\cdot, \cdot)$, which is reflexive, symmetric, and transitive and should capture semantic equivalence between input texts. In practice, $E$ is computed using bi-directional entailment predictions from a Natural Language Inference (NLI) model, such as DeBERTa [22] or a prompted LLM, that classifies relations between pairs of texts as "entailment," "neutral," or "contradiction". Two texts are semantically equivalent if they entail each other bi-directionally. Semantic clusters are obtained by greedily aggregating generations into clusters of equivalent meaning. We can now define semantic entropy.

**Definition 2.2.** *For an input $x$ and semantic clusters $C \in \Omega$, where $\Omega$ is a set of all semantic clusters, Semantic Entropy (SE) is defined as*

$$\text{SE}(x) = -\sum_{C \in \Omega} p(C \mid x) \log p(C \mid x) = -\sum_{C \in \Omega} \left( \left( \sum_{s \in C} p(s \mid x) \right) \log \left[ \sum_{s \in C} p(s \mid x) \right] \right). \tag{4}$$

In practice, it is not possible to calculate $\sum_C p(C \mid x) \log p(C \mid x)$ because of the intractable number of semantic clusters. Instead, SE uses a Rao-Blackwellized Monte Carlo estimator

$$\text{SE}(x) \approx -\sum_{i=1}^M p'(C_i|x) \log p'(C_i|x), \tag{5}$$

where $C_i$ are $M$ clusters extracted from the generations and $p'(C_i \mid x)$ is a normalized semantic probability, $p'(C_i \mid x) = p(C_i|x)/\sum_i p(C_i|x)$, which we refer to as $p(C_i|x)$ in the following for simplicity. SE can be extended to cases where token likelihoods are not available by approximating $p(C_i|x)$ with the fraction of generated texts in each cluster, $p(C_i|x) \approx \sum_{i=1}^N \mathbb{I}(S_i \in C_i)/N$. We refer to this variant as *Discrete Semantic Entropy* [16].

## 3 Kernel Language Entropy

This section introduces Kernel Language Entropy (KLE), our novel approach to computing semantic uncertainty that accounts for fine-grained similarities between generations for better uncertainty quantification. We introduce two variants of KLE: the first, simply called KLE, operates directly on the generated texts, and the second, KLE-c operates on the space of semantic clusters.

**Motivating Example.** Figure 1 illustrates the advantages of KLE (to be precise, the KLE-c variant) over other methods such as SE. Imagine querying two LLMs such that the outputs of $LLM_1$ are all semantically different and those of $LLM_2$ are semantically similar *but not equivalent*. For simplicity, we assume an equal amount of clusters between LLMs and equal likelihoods of clusters $p(C_i|\text{inp}) = p(C_i'|\text{inp})$. SE would not distinguish between those cases and, thus, would misleadingly predict equal uncertainty. KLE on the other hand, will correctly assign lower uncertainty to the outputs of $LLM_2$, its kernels accounting for the fact that $LLM_2$ produces semantically similar outputs.

Before introducing KLE, we recall the definition of a positive semidefinite (PSD) kernel.

**Definition 3.1.** *For a set $\mathcal{X} \neq \emptyset$, a symmetric function $K : \mathcal{X} \times \mathcal{X} \to \mathbb{R}$ is called a PSD kernel if for all $n > 0, x_i \in \mathcal{X}, \alpha_i \in \mathbb{R}$*

$$\sum_{i=1}^{n} \sum_{j=1}^{n} \alpha_i \alpha_j K(x_i, x_j) \geq 0. \tag{6}$$

*For a finite set $\mathcal{X}$, a PSD kernel is a PSD matrix of the size $|\mathcal{X}|$.*

## 3.1 Semantic Kernels and KLE

Next, we define ***semantic kernels***, denoted $K_{\text{sem}}$, as unit trace[1] positive semidefinite kernels over the finite domain of *generated* texts. Unit trace PSD matrices are also called density matrices. These kernels should, informally speaking, capture the semantic similarity[2] between the texts such that $K(s_1, t_1) > K(s_2, t_2)$ if and only if texts $s_1$ and $t_1$ are more semantically related than texts $s_2$ and $t_2$. Analogously, we define semantic kernels over semantic clusters of texts, in which case the kernel should capture the semantic similarity between the clusters. In practice, there are multiple ways to concretely specify a proper semantic kernel, and some options are described in Section 3.2.

**The von Neumann Entropy.** We propose to use the von Neumann entropy (VNE) to evaluate the uncertainty associated with a semantic kernel.

**Definition 3.2** (Von Neumann Entropy). *For a unit trace positive semidefinite matrix $A \in \mathbb{R}^{n \times n}$, the von Neumann entropy (VNE; [72]) is defined as*

$$\text{VNE}(A) = -\text{Tr}[A \log A]. \tag{7}$$

It can be shown that $\text{VNE}(A) = \sum_{i}^{n} -\lambda_i \log \lambda_i$ where $\lambda_i, 1 \leq i \leq n$ are the eigenvalues of $A$. Within this definition, we assume $0 \log 0 = 0$. This reformulation shows that VNE is, in fact, the Shannon entropy over the eigenvalues of a kernel.

**Kernel Language Entropy (KLE).** We can now define Kernel Language Entropy, as the VNE of a semantic kernel.

**Definition 3.3** (Kernel Language Entropy). *Given a set of LLM generations $S_1, \ldots, S_N$, an input $x$, and semantic kernel $K_{sem}$ over these generations and input, we define **Kernel Language Entropy** (KLE) as the von Neumann entropy of a semantic kernel $K_{sem}$:*

$$\text{KLE}(x) = \text{VNE}(K_{sem}). \tag{8}$$

The von Neumann entropy has the following properties, which are aligned with the overarching goal of measuring the uncertainty of a set of generations.

**Proposition 3.4** (Properties of the von Neumann Entropy [5]). *The VNE of a unit trace positive semidefinite kernel has the following properties:*

1. *The VNE of a kernel with only one non-zero element is equal to 0.*

2. *The VNE is invariant under changes of basis $U$: $\text{VNE}(K) = \text{VNE}(UKU^{\top})$.*

3. *The VNE is concave. For a set of positive coefficients $\alpha_i$, $\sum_{i=1}^{k} \alpha_i = 1$, and density matrices $K_i$, it holds that $\text{VNE}\left(\sum_{i=1}^{k} \alpha_i K_i\right) \geq \sum_{i=1}^{k} \alpha_i \text{VNE}(K_i)$.*

---

[1]Kernels with $\text{Tr}[K] = 1$ are called *unit trace kernels*.
[2]Or more broadly semantic *relatedness*, including antonymy, meronymy, as well as semantic similarity [7].

Let us briefly discuss the practical implications of these properties. **Property 1** states that if an LLM outputs a single answer (for KLE) or a semantic cluster (for KLE-c), the VNE is zero, indicating high certainty. **Property 2** is significant as it allows the VNE to be calculated in practice as the Shannon entropy of the diagonal elements of an orthogonalized kernel, which can be interpreted as a disentangled representation of a semantic kernel. **Property 3** states that entropy is concave, meaning that the entropy of a combined system is greater than or equal to the entropy of its individual parts, a common requirement for entropy metrics. The intuition behind our use of the VNE for LLMs also relates to its origins in quantum information theory.

**The VNE in Quantum Information Theory.** In quantum information theory, the states of a quantum system (or pure states) are defined as unit vectors in $\mathbb{C}^N$. However, experiments often result in statistical mixtures of pure quantum states, represented as density matrices. The VNE is used to evaluate the entropy of the mixed states. Analogously, we can think of KLE as considering each answer as a mixture of pure "semantic meanings", measuring the entropy of this mixture. We refer the reader to Aaronson [1] for further background reading on the VNE and quantum information theory.

---

**Algorithm 1** Kernel Language Entropy

**Require:** LLM, Input $x \in \mathcal{T}^L$, Number of samples $n$, Boolean kle-c indicating variant, Semantic kernels $K_i$
1: Initialize a *multiset* of answers $\mathcal{O} \leftarrow \emptyset$
2: **for** $k \leftarrow 1$ to $n$ **do**     ▷ Sampling $n$ answers
3:     Add LLM$(x)$ to $\mathcal{O}$
4: **end for**
5: **if** kle-c **then**
6:     Update $\mathcal{O} \leftarrow$ cluster$(\mathcal{O})$     ▷ as in [36]
7: **end if**
8: Combine $K_i(\mathcal{O}, \mathcal{O})$ in $K_{\text{sem}}$     ▷ see Sec. 3.2
9: Return VNE$(K_{\text{sem}})$     ▷ Eq. (8)

---

**KLE-c.** Instead of defining semantic kernels directly over individual model generations, we can also apply KLE to clusters of semantic equivalence. We call this variant of our method KLE-c. Although KLE is more general than KLE-c for non-trivial clusterings, KLE-c can provide practical value as it is cheaper to compute and more interpretable due to its smaller kernel sizes.

**Algorithm.** Algorithm 1 provides a generic description of the steps required to compute KLE. We describe the practical details for defining and combining semantic kernels later in Sec. 3.2.

**Computational Complexity.** The computational complexity of KLE is approximately identical to SE which requires sampling from an LLM $N$ times and running the entailment model $O(N^2)$ times. Additionally, KLE requires $O(N^3)$ elementary operations for kernel and VNE calculation. The actual cost of this is negligible in comparison to the forward passes through the LLM or entailment model.

## 3.2 Semantic Graph Kernels

This section describes a practical approach for constructing semantic kernels over LLM generations or semantic clusters. Concretely, we apply NLI models to construct *semantic graphs* over the LLM outputs and then borrow from graph kernel theory to construct kernels from these graphs. A similar notion of semantic graphs derived from NLI models was proposed by Lin et al. [44] for black-box LLM uncertainty quantification.

**Graph Theory Preliminaries.** First, let us recall the basics of graph theory. A graph is a pair of two sets $G = (V, E)$, where $V = \{1, \ldots, n\}$ is a set of $n$ vertices and $E \subseteq V \times V$ is a set of edges. A graph is called weighted when a weight is assigned to each edge, and the weight matrix $W_{ij}$ contains weights between nodes $i$ and $j$. For unweighted graphs, we can use a binary adjacency matrix to encode edges between nodes. The degree matrix $D$ is a diagonal $|V| \times |V|$ matrix with $D_{ii} = \sum_{j=1}^{|V|} W_{ij}$. The *graph Laplacian* is defined as $L = D - W$. $L$ is a positive semidefinite matrix, and eigenvalues of $L$ are often used to study the structure of graphs [10, 71].

**Semantic Graph.** We define semantic graphs as graphs over LLM generations ($G_{\text{sem}}$) or semantic clusters ($G_{\text{sem-c}}$). For $G_{\text{sem}}$, edges can be defined as a function of NLI predictions in both directions: $W_{ij} = f(\text{NLI}(S_i, S_j), \text{NLI}(S_j, S_i))$, where NLI are the predicted probabilities for *entailment*, *neutral*, and *contradiction* for $S_i$ and $S_j$. For example, $f$ could be the weighted sum over the predicted probabilities for entailment and neutral classes. For $G_{\text{semc-c}}$, the weights between the clusters are computed by summing the entailment predictions over the generations assigned to the clusters, $W_{ij} = \sum_{s \in C_i} \sum_{t \in C_j} f(\text{NLI}(s, t), \text{NLI}(t, s))$.

**Graph Kernels.** When a semantic graph is obtained, KLE calculates graph kernels over semantic graph nodes to compute a distance measure. Since graphs are discrete and finite, any positive

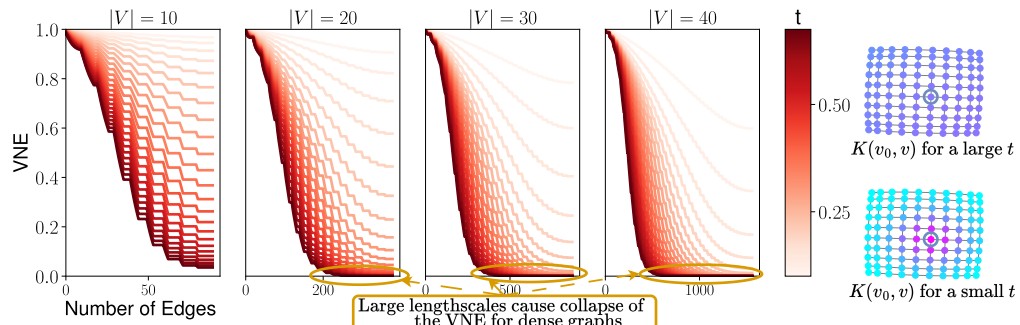

Figure 2: Entropy Convergence Plots for heat kernels. For graphs of various sizes $|V|$, we grow the number of edges and examine the VNE. For large lengthscales $t$, corresponding to darker colored curves, the VNE quickly converges to zero. We can use these plots to determine kernel hyperparameters without validation sets. The VNE is scaled to start at 1 for visualization purposes.

semidefinite matrix would be a kernel over the graph. However, we seek kernels that exploit knowledge about the graph structure. We, therefore, adopt Partial Differential Equation (PDE) and Stochastic Partial Differential Equation (SPDE) approaches to graph kernels [34, 6, 57]. If $u \in \mathbb{R}^n$ is a signal over the nodes of a graph, the **heat kernel** is a solution to the partial differential equation $\partial u / \partial t + Lu = 0$ and the **Matérn kernel** is a solution to the stochastic differential equation, $(2\nu/\kappa^2 + L)^{\frac{\nu}{2}} u = w$, where $w$ is white noise over the graph nodes and $L$ is the graph Laplacian defined above. The corresponding solutions to these equations are:

$$K_t = e^{-tL} \quad \text{[HEAT]} \qquad\qquad K_{\nu\kappa} = (2\nu/\kappa^2 I + L)^{-\nu} \quad \text{[MATÉRN]}. \qquad (9)$$

These kernels allow for the incorporation of a distance measure that reflects the graph's locality properties (right part of Fig. 2). For example, the Taylor series of the heat kernel can be shown to be equal to a sum of powers of random walk matrices. Both kernels have hyperparameters: lengthscales $t$ in the heat kernel, $\kappa$ in Matérn kernels, and $\nu$ in the Matérn kernel, often interpreted as smoothness. The scaled eigenvalues of the Matérn kernel converge to the eigenvalues of the heat kernel [6] when $\nu$ goes to infinity. Matérn kernels provide more flexibility at the cost of the additional parameter. Note that any kernel can be normalized into a unit trace kernel via $K(x, y) \leftarrow K(x, y)(K(x, x)K(y, y))^{-1/2}/N$, where $N$ is the size of $K$. We refer to [34, 57, 6] for further background reading on graph kernels.

**Kernel Hyperparameters.** We propose two ways to select the hyperparameters of the heat and Matérn kernels: either by maximizing the validation set performance or by selecting parameters from what we call *Entropy Convergence Plots*, illustrated in Fig. 2. We obtain these plots by defining a set of progressively denser graphs $G_1 \prec \ldots \prec G_K$. These can be obtained by starting from a graph without edges and a fixed number of vertices and adding new edges either randomly or by filling in the adjacencies of each node sequentially. We then plot the VNE against the number of edges in the graphs $G_i$. We analyze the von Neumann entropy over these plots to avoid pathologies connected to the fact that for large lengthscales, the VNE converges rather quickly, and such behavior should generally be avoided. For all remaining values, we can either choose hyperparameters randomly from the range of non-collapsing hyperparameters or rely on prior domain knowledge.

**Kernel Combination.** KLE offers the additional flexibility of combining kernels from various methods (e.g., multiple NLI models, different graph kernels, or other methods). For example, we can combine multiple kernels using convex combinations, $K = \sum_{i=1}^{P} \alpha_i K_i$, where $\sum_{i=1}^{P} \alpha_i = 1$.

### 3.3 Kernel Language Entropy Generalizes Semantic Entropy

The semantic kernels used in KLE are more informative than the semantic equivalence relations used in SE [36]. The next theorem shows that KLE can recover SE for any semantic clustering.

**Theorem 3.5** (KLE and KLE-c generalize SE). *For any semantic clustering, there exists a semantic kernel over texts $K_{sem}(s, s')$ such that the VNE of this kernel is equal to semantic entropy (computed as in Eq. (5)). Moreover, there exists a semantic kernel over clusters $K_{sem}(c, c')$ such that the VNE of this kernel is equal to SE.*

*Proof Sketch.* For any semantic clustering, we consider a kernel with a block diagonal structure. Each block corresponds to a semantic cluster, and cluster likelihoods are normalized by the size of the cluster, $p(C_i|x)/m_i$. This is a valid semantic kernel and the KLE for this kernel equals the SE. Thm. B.1 and Thm. B.2 in the Appendix contain the detailed proofs. □

The proof of Thm. 3.5 shows that the block diagonal semantic kernels used with KLE can recover semantic entropy for any clustering. However, there are other kernels available that allow KLE to be more expressive than SE. Comparing KLE and KLE-c, we find that KLE is more general than KLE-c for any non-trivial clustering.

## 4    Related Work

In the context of machine learning, the VNE has been studied theoretically[4], applied to GAN regularization [33], and the exponential of the VNE has been used for effective rank and sample diversity analysis [64, 19].

The first attempts at estimating the entropy of language date back to the 1950s [65], and today, techniques for uncertainty quantification are widely used in natural language processing. For instance, Desai and Durrett [15] and Jiang et al. [28] presented calibration techniques for classification tasks. Xiao and Wang [73] empirically showed that, for various tasks, including sentiment analysis and named entity recognition, measuring model uncertainty can be used to improve performance. Calibration techniques have also been applied in machine translation tasks to improve accuracy [37].

Malinin and Gales [50] discussed the challenges of estimating uncertainty in sequential models. Several previous works have queried LLMs to elicit statements about uncertainty, either via fine-tuning or by directly including previous LLM generations in the prompt [30, 9, 54, 43, 53, 21, 63, 68, 12, 74, 36]. Zhang et al. [76] studied UQ for long text generation. Quach et al. [61] used conformal predictions to quantify LLM uncertainty, which is orthogonal to the approach we pursue here. Yang et al. [75] have shown that Bayesian modeling of LLMs using low-rank Laplace approximations improves calibration in small-scale multiple-choice settings. Lin et al. [44] extended the work of Kuhn et al. [36] on semantic entropy by introducing the use of the Laplacian of semantic graphs and applying spectral graph analysis for UQ in black-box LLMs. Aichberger et al. [2] proposed a new method for sampling diverse answers from LLMs, and Liu et al. [47] proposed improving calibration by adding an extra linear layer; more diverse sampling strategies and better calibration could improve KLE as well.

There are a variety of ways besides model uncertainty to detect hallucinations in LLMs such as querying external knowledge bases [17, 42, 70], hidden state interventions [77, 24, 46], using probes [8, 41, 48], or applying fine-tuning [31, 67]. KLE is complementary to many of these directions and focuses on estimating more fine-grained semantic uncertainty. It can either be used to improve these approaches or be combined with them sequentially.

## 5    Experiments

**Datasets and Models.** Our experiments span over 60 dataset-model pairs. We evaluate our method on the following tasks covering different domains of natural language generation: general knowledge (TriviaQA [29] and SQuAD [62]), biology and medicine (BioASQ [35]), general domain questions from Google search (Natural Questions, NQ [38]), and natural language math problems (SVAMP [60]). We generally discard the context associated with each input for all datasets except SVAMP, as the tasks become too easy for the current generation of models when context is provided. We use the following LLMs: Llama-2 7B, 13B, and 70B [69], Falcon 7B and 40B [3], and Mistral 7B [27], using both standard and instruction-tuned versions of these models. As the NLI model for defining semantic graphs or semantic clusters, we use DeBERTa-Large-MNLI [22].

**Baselines.** As baseline methods, we compare KLE with semantic entropy [36], discrete semantic entropy [16, 36], token predictive entropy [50], embedding regression [16], and P(True) [30]. For embedding regression, we train a logistic regression model on the last layer's hidden states to predict whether a given LLM answer is correct.

**KLE Kernels.** We propose to use the following two semantic kernels with KLE: $K_{\text{HEAT}}$ and $K_{\text{FULL}}$. Both are obtained from the weighted graph $W_{ij} = w\,\text{NLI}'(S_i, S_j) + w\,\text{NLI}'(S_j, S_i)$, where $w = (1, 0.5, 0)^\top$ is a weight vector. Here, we assume that $\text{NLI}'$ returns a one-hot prediction over

Table 1: Detailed experimental results for Llama 2 70B Chat and Falcon 40B Instruct.

| | Method | BioASQ [35] | | NQ [38] | | SQuAD [62] | | SVAMP [60] | | Trivia QA [29] | |
|---|---|---|---|---|---|---|---|---|---|---|---|
| | | AUROC | AUARC | AUROC | AUARC | AUROC | AUARC | AUROC | AUARC | AUROC | AUARC |
| Llama 2 70B Chat | SE [36] | $0.74 \pm 0.04$ | $0.90 \pm 0.01$ | $0.71 \pm 0.03$ | $0.47 \pm 0.03$ | $0.66 \pm 0.03$ | $0.65 \pm 0.03$ | $0.62 \pm 0.03$ | $0.61 \pm 0.03$ | $0.77 \pm 0.03$ | $0.79 \pm 0.02$ |
| | DSE [36] | $0.75 \pm 0.04$ | $0.90 \pm 0.01$ | $0.71 \pm 0.03$ | $0.46 \pm 0.03$ | $0.66 \pm 0.03$ | $0.65 \pm 0.03$ | $0.63 \pm 0.03$ | $0.61 \pm 0.03$ | $0.77 \pm 0.03$ | $0.79 \pm 0.02$ |
| | PE [50] | $0.69 \pm 0.04$ | $0.90 \pm 0.01$ | $0.67 \pm 0.03$ | $0.44 \pm 0.03$ | $0.65 \pm 0.03$ | $0.65 \pm 0.03$ | $0.59 \pm 0.03$ | $0.58 \pm 0.03$ | $0.61 \pm 0.03$ | $0.73 \pm 0.03$ |
| | P(True) [30] | $0.86 \pm 0.03$ | $\mathbf{0.92} \pm 0.01$ | $\mathbf{0.78} \pm 0.03$ | $0.50 \pm 0.03$ | $0.69 \pm 0.03$ | $\mathbf{0.68} \pm 0.03$ | $0.74 \pm 0.02$ | $0.68 \pm 0.03$ | $0.76 \pm 0.03$ | $0.79 \pm 0.02$ |
| | ER | $0.70 \pm 0.05$ | $0.89 \pm 0.01$ | $0.58 \pm 0.03$ | $0.39 \pm 0.03$ | $0.63 \pm 0.03$ | $0.64 \pm 0.03$ | $0.68 \pm 0.03$ | $0.64 \pm 0.03$ | $0.76 \pm 0.03$ | $0.79 \pm 0.02$ |
| | KLE($K_{\mathrm{HEAT}}$) | $0.87 \pm 0.03$ | $\mathbf{0.92} \pm 0.01$ | $\mathbf{0.78} \pm 0.02$ | $\mathbf{0.51} \pm 0.03$ | $\mathbf{0.71} \pm 0.03$ | $\mathbf{0.68} \pm 0.03$ | $\mathbf{0.76} \pm 0.02$ | $\mathbf{0.69} \pm 0.03$ | $\mathbf{0.84} \pm 0.03$ | $\mathbf{0.82} \pm 0.02$ |
| | KLE($K_{\mathrm{FULL}}$) | $\mathbf{0.88} \pm 0.03$ | $\mathbf{0.92} \pm 0.01$ | $0.77 \pm 0.02$ | $0.50 \pm 0.03$ | $0.70 \pm 0.03$ | $\mathbf{0.68} \pm 0.03$ | $0.70 \pm 0.03$ | $0.65 \pm 0.03$ | $0.80 \pm 0.03$ | $0.81 \pm 0.02$ |
| Falcon 40B Instr | SE [36] | $0.85 \pm 0.02$ | $0.90 \pm 0.01$ | $\mathbf{0.78} \pm 0.03$ | $\mathbf{0.43} \pm 0.03$ | $0.66 \pm 0.03$ | $0.63 \pm 0.03$ | $0.66 \pm 0.03$ | $0.63 \pm 0.03$ | $0.79 \pm 0.03$ | $0.72 \pm 0.03$ |
| | DSE [36] | $0.85 \pm 0.02$ | $0.89 \pm 0.01$ | $0.77 \pm 0.03$ | $0.40 \pm 0.03$ | $0.66 \pm 0.03$ | $0.62 \pm 0.03$ | $0.67 \pm 0.03$ | $0.61 \pm 0.03$ | $0.79 \pm 0.03$ | $0.71 \pm 0.03$ |
| | PE [50] | $0.75 \pm 0.03$ | $0.87 \pm 0.01$ | $0.71 \pm 0.03$ | $0.38 \pm 0.03$ | $0.63 \pm 0.03$ | $0.60 \pm 0.03$ | $0.59 \pm 0.03$ | $0.57 \pm 0.03$ | $0.68 \pm 0.03$ | $0.66 \pm 0.03$ |
| | P(True) [30] | $0.87 \pm 0.03$ | $0.89 \pm 0.01$ | $0.71 \pm 0.03$ | $0.37 \pm 0.03$ | $0.66 \pm 0.03$ | $0.61 \pm 0.03$ | $0.73 \pm 0.03$ | $0.67 \pm 0.03$ | $0.72 \pm 0.03$ | $0.69 \pm 0.03$ |
| | ER | $0.74 \pm 0.04$ | $0.85 \pm 0.02$ | $0.73 \pm 0.03$ | $0.39 \pm 0.03$ | $0.63 \pm 0.03$ | $0.61 \pm 0.03$ | $0.68 \pm 0.03$ | $\mathbf{0.68} \pm 0.03$ | $0.76 \pm 0.03$ | $0.69 \pm 0.03$ |
| | KLE($K_{\mathrm{HEAT}}$) | $\mathbf{0.92} \pm 0.01$ | $\mathbf{0.91} \pm 0.01$ | $0.76 \pm 0.03$ | $0.42 \pm 0.03$ | $\mathbf{0.70} \pm 0.03$ | $\mathbf{0.66} \pm 0.03$ | $\mathbf{0.77} \pm 0.02$ | $\mathbf{0.68} \pm 0.03$ | $\mathbf{0.80} \pm 0.02$ | $\mathbf{0.74} \pm 0.03$ |
| | KLE($K_{\mathrm{FULL}}$) | $0.90 \pm 0.02$ | $\mathbf{0.91} \pm 0.01$ | $\mathbf{0.78} \pm 0.03$ | $\mathbf{0.43} \pm 0.03$ | $0.69 \pm 0.03$ | $0.65 \pm 0.03$ | $0.69 \pm 0.03$ | $0.64 \pm 0.03$ | $\mathbf{0.80} \pm 0.03$ | $0.73 \pm 0.03$ |

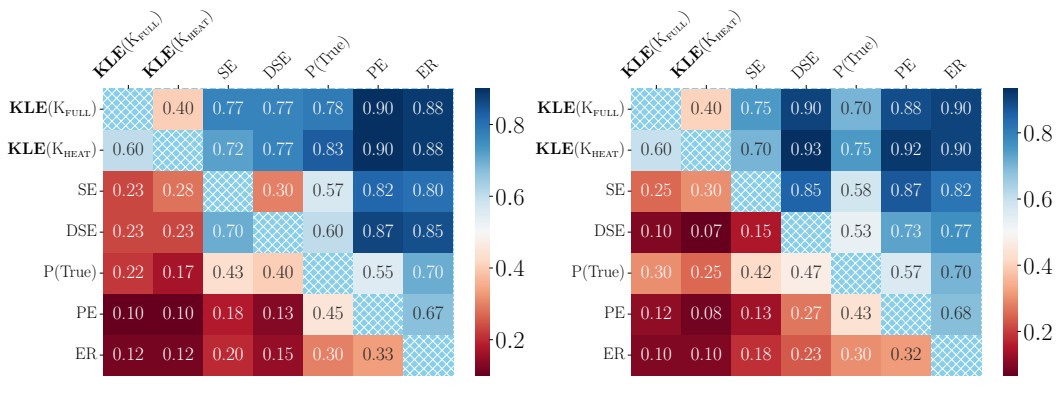

(a) Win rate measured with AUROC  (b) Win rate measured with AUARC

Figure 3: Summary of **60** experimental scenarios. Each cell contains the fraction of experiments where a method from a row outperforms a method from a column. Our methods are labeled KLE($\cdot$). Values larger than or equal to $0.62$ and less than or equal to $0.38$ correspond to the significance level $p < 0.05$ according to the binomial statistical significance test.

(entailment, neutral class, contradiction). $K_{\mathrm{HEAT}}$ is a heat kernel over this graph. We further propose $K_{\mathrm{FULL}} = \alpha K_{\mathrm{HEAT}} + (1 - \alpha)K_{\mathrm{SE}}$, where $\alpha \in [0, 1]$ and $K_{\mathrm{SE}}$ is a semantic entropy kernel. We ablate these kernel choices in our experiments below.

**Evaluation metrics.** Following previous work, we evaluate uncertainty methods by measuring their ability to predict the correctness of model responses, calculating the Area under the Receiver Operating Curve (AUROC). Further, uncertainty metrics can be used to refuse answering when uncertainty is high, increasing model accuracy on the subset of questions with uncertainty below a threshold. We measure this with the Area Under the Accuracy-Rejection Curve (AUARC, [56]). The rejection accuracy at a given uncertainty threshold is the accuracy of the model on the subset of inputs for which uncertainty is lower than the threshold; the AUARC score computes the area under the rejection accuracy curve for all possible thresholds.

**Sampling.** We sample 10 answers per input via top-K sampling with $K = 50$ and nucleus sampling with $p = 0.9$ at temperature $T = 1$. To assess model accuracy, we draw an additional low-temperature sample ($T = 0.1$) and ask an additional LLM (Llama 3 8B Instruct) to compare the model response to the ground truth answer provided by the datasets. We evaluated the accuracy-checking performance of Llama 3 8B Instruct by comparing its assessments with human raters, finding 90% agreement across 100 cases. We also compared its evaluations with GPT-4 evaluations on the TriviaQA dataset, using answers generated by Llama-2-70B-chat, and observed a 95% agreement.

**Statistical significance.** We assess statistical significance in two ways. First, we run a large number of experimental scenarios (60 model-dataset pairs), and second, for each experimental scenario, we also obtain confidence intervals over 1000 bootstrap resamples. We note that standard errors in each scenario are more representative of the LLM and the dataset rather than the method. Therefore, our main criterion for comparing the methods is based on the fraction of experimental cases where our method outperforms baselines (assessed with a binomial statistical significance test).

**KLE outperforms previous methods.** We compare the performance of UQ methods over 60 scenarios (12 models, five datasets). Figure 3 shows the heatmaps of pairwise win rates. We observe that both our methods, $\text{KLE}(K_{\text{HEAT}})$ and $\text{KLE}(K_{\text{FULL}})$, are superior to the baselines. Furthermore, Table 1 shows the detailed results for the two largest models from our experiments, Llama 2 70B Chat and Falcon 40B Instruct. The results show that for the largest models, our method consistently achieves best results compared to baselines. In Fig. D.3 and Fig. D.4, we show the experimental results for all considered models. Importantly, our best method, $\text{KLE}(K_{\text{HEAT}})$, does not require token-level probabilities from a model and works in black-box scenarios.

**KLE hyperparameters can be selected without validation sets.** We compare the strategies of hyperparameter selection from Sec. 3.2: entropy convergence plots and validation sets (100 samples per dataset except for SVAMP, where we used default hyperparameters). We observe that default hyperparameters achieve similar results as selecting hyperparameters from validation sets and conclude that choosing default hyperparameters from entropy convergence plots is a good way to select hyperparameters in practice. In Fig. 4, we compare the two strategies for selecting hyperparameters, and see that the ranking of the methods remains stable and the pairwise win-rates are similar for both methods.

**Many design choices outperform existing methods, the best is $\text{KLE}(K_{\text{HEAT}})$.** Next, in Fig. 4, we compare several design choices for KLE: choosing a kernel (heat or Matérn), using KLE-c, combining kernels via a weighted sum or product, and using the probabilities returned by DeBERTa for edge weights. The superscript indicates the type of a graph: no superscript indicates a weighted graph as described above, DB means weights are assigned using probabilities from DeBERTa, and C means a weighted graph over clusters (KLE-c). The subscript indicates the semantic kernels: SE stands for a diagonal kernel with semantic probabilities ($K_{\text{SE}}$), HEAT and MATÉRN for the type of kernel ($K_{\text{HEAT}}$ and $K_{\text{MATÉRN}}$), and $\star$ for the best of Heat and Matérn kernels. We observe that even though all design choices outperform SE, the heat kernel over a weighted semantic graph, $\text{KLE}(K_{\text{HEAT}})$, was overall the best. Additionally, we notice that the methods based on token likelihoods are performing better for non-instruction-tuned models, and we can practically recommend including semantic probabilities (e.g., use variations of $K_{\text{FULL}}$) if KLE is used in non-instruction-tuned scenarios (see Fig. D.5).

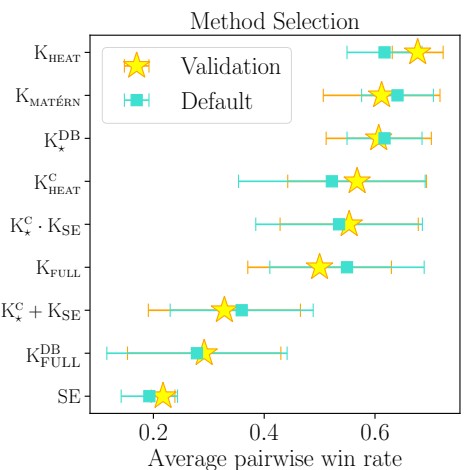

Figure 4: Comparison of various design choices for semantic graph kernels. ⭐ represents the best hyperparameters and 🟩 – defaults. Error bars are twice the standard error. Summary of 48 experiments. KLE consistently outperformed SE across all the kernels evaluated.

**KLE is better in practice because it captures more fine-grained semantic relations than SE.** The performance of KLE improves over SE because in complex free-form language generation scenarios, such as those studied here, LLMs can generate similar but not strictly equal answers. SE assigns these to separate clusters, predicting high entropy. By contrast, our method can account for semantic similarities using the kernel metric in the space of meanings over generated texts, and predict reduced uncertainty if necessary. We give a detailed illustrative example for which KLE provides better uncertainty estimates than SE from the NQ Open dataset in Fig. C.2.

## 6 Discussion

Measuring semantic uncertainty in LLMs is a challenging and important problem. It requires navigating the semantic space of the answers, and we have suggested a method, KLE, that encodes a similarity measure in this space via semantic kernels. KLE allows for fine-grained estimation of uncertainty and is an expressive generalization of semantic entropy. We provided several specific design choices by defining NLI-based semantic graphs and kernels, and studying kernel hyperparameters. We have evaluated KLE across various domains of natural language generation, and it has demonstrated superior performance compared to the previous methods. Our method works both for

white- and black-box settings, enabling its application to a wide variety of practical scenarios. We hope to inspire more work that moves from semantic *equivalence* to semantic *similarity* for estimating semantic uncertainty in LLMs.

**Broader Impact.** Our work advances the progress toward safer and more reliable uses of LLMs. KLE can positively impact areas that involve using LLMs by providing more accurate uncertainty estimates, which can filter out a proportion of erroneous outputs.

**Limitations.** One limitation of the proposed method is that it requires multiple samples from an LLM, which generally increases the generation cost. However, in safety-critical tasks, the potential cost of hallucination should outweigh the cost of sampling multiple answers, so reliable uncertainty quantification via KLE should always be worthwhile. Additionally, we study semantic kernels derived from NLI-based semantic graphs, but other semantic kernels warrant investigation, such as kernels on embeddings. Moreover, the NLG landscape is highly diverse, and the method should be carefully evaluated for other potential applications of LLMs, such as code generation. Lastly, our method estimates uncertainty using predictive entropy, as commonly done in Bayesian deep learning. However, in applications where confidence estimates are important, alternative methods should be considered.

## Acknowledgments and Disclosure of Funding

This work was supported by the Research Council of Finland (Flagship programme: Finnish Center for Artificial Intelligence FCAI, and grants 352986, 358246) and EU (H2020 grant 101016775 and NextGenerationEU). We also acknowledge the computational resources provided by the Aalto Science-IT Project from Computer Science IT. The authors wish to acknowledge CSC – IT Center for Science, Finland, for computational resources.

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

# Supplementary Material:
# Kernel Language Entropy: Fine-grained Uncertainty Quantification for LLMs from Semantic Similarities

## A   Background

### A.1   Linear Algebra

**Definition A.1.** *For a set $\mathcal{X} \neq \emptyset$, a symmetric function $K : \mathcal{X} \times \mathcal{X} \to \mathrm{R}$ is called a positive-semidefinite kernel if for all $n > 0, x_i \in \mathcal{X}, \alpha_i \in \mathbb{R}$*

$$\sum_{i=1}^{n} \sum_{j=1}^{n} \alpha_i \alpha_j K(x_i, x_j) \geq 0. \tag{A.1}$$

*For a finite set $\mathcal{X}$, a positive semidefinite kernel is a positive semidefinite matrix of the size $|\mathcal{X}|$.*

**Lemma A.2.** *For a block diagonal matrix*

$$A = \begin{pmatrix} A_{11} & 0 & 0 & \dots & 0 \\ 0 & A_{22} & 0 & \dots & 0 \\ 0 & 0 & A_{33} & \dots & 0 \\ \vdots & \vdots & \vdots & \ddots & \vdots \\ 0 & 0 & 0 & \dots & A_{nn} \end{pmatrix}$$

*eigenvalues are all eigenvalues of the blocks $A_{ii}$ combined, or equivalently $\det(A - \lambda I) = 0 \Leftrightarrow \prod_{i=1}^{n} \det(A_{ii} - \lambda I) = 0$*

*Proof.* Notice, that a block diagonal matrix can be decomposed into the following product:

$$A = \begin{pmatrix} A_{11} & 0 & \dots & 0 \\ 0 & I_{22} & \dots & 0 \\ \vdots & \vdots & \ddots & \vdots \\ 0 & 0 & \dots & I_{nn} \end{pmatrix} \begin{pmatrix} I_{11} & 0 & \dots & 0 \\ 0 & A_{22} & \dots & 0 \\ \vdots & \vdots & \ddots & \vdots \\ 0 & 0 & \dots & I_{nn} \end{pmatrix} \cdots \begin{pmatrix} I_{11} & 0 & \dots & 0 \\ 0 & I_{22} & \dots & 0 \\ \vdots & \vdots & \ddots & \vdots \\ 0 & 0 & \dots & A_{nn} \end{pmatrix},$$

where $I_{ii}$ are the identity matrices of the same size as $A_{ii}$.

By using the product rule for determinants, we obtain $\det(A - \lambda I) = 0 \Leftrightarrow \prod_{i=1}^{n} \det(A_{ii} - \lambda I) = 0$. $\square$

**Lemma A.3** (Horn and Johnson [25]). *An all-ones matrix $J$ of size $n$ has eigenvalues $\{n, \underbrace{0, \dots, 0}_{n-1}\}$.*

### A.2   Discrete Mathematics

Throughout the text, we often refer to the notion of equivalence relation. We remind readers of the definition of equivalence relation here.

**Definition A.4.** *Equivalence relation is a binary relation $E(\cdot, \cdot)$ on a set $\mathcal{X}$, that is for any $x, y, z \in \mathcal{X}$, this relation is*

1. *reflexive $E(x, x)$,*

2. *symmetric $E(x, y) \iff E(y, x)$,*

3. *transitive if $E(x, y)$ and $E(y, z)$ then $E(x, z)$.*

# B    Theoretical Results and Proofs

In this section, we prove Thm. 3.5, for convenience we separate it into two theorems for KLE and KLE-c.

**Theorem B.1** (KLE is a generalization of SE). *For any semantic clustering, there exists a semantic kernel over texts $K_{sem}(s, s')$ such that the VNE of this kernel is equal to semantic entropy (computed as in Eq. (5)).*

*Proof.* Let us fix an arbitrary semantic clustering over $M$ clusters $\mathcal{C} = \{C_1, \ldots, C_M\}$, with the size of each cluster $m_i$. Now, we will construct a kernel $K$ for an input $x$ such that the von Neumann entropy with this kernel will be equal to the semantic entropy of the texts $\text{VNE}(K) = \text{SE}(x, \mathcal{C})$. Let us consider a block-diagonal kernel $K$. We will denote blocks of $K$ as $K_1, \ldots, K_M$:

$$K = \begin{pmatrix} K_1 & 0 & 0 & \ldots & 0 \\ 0 & K_2 & 0 & \ldots & 0 \\ 0 & 0 & K_3 & \ldots & 0 \\ \vdots & \vdots & \vdots & \ddots & \vdots \\ 0 & 0 & 0 & \ldots & K_M \end{pmatrix} \tag{B.1}$$

where $M$ corresponds to the number of semantic clusters. The size of each block $K_i$ is $m_i \times m_i$. Note that because $K$ is block-diagonal, it follows that $\text{VNE}(K) = \sum_{i=1}^{M} \text{VNE}(K_i)$. Consequently, if

1. $\text{VNE}(K_i) = -p(C_i|x) \log p(C_i|x)$,

2. the sum of eigenvalues of $K_i$ is equal to $p(C_i|x)$,

3. $K$ is positive semidefinite and unit trace,

then $\text{VNE}(K) = \text{SE}(s|x)$.

Let us define each block as $K_i = \frac{p(C_i|x)}{m_i} J_{m_i}$ where $J_{m_i}$ is an all-ones matrix of size $m_i \times m_i$.

Next, we prove that the desired properties from the list above hold. Indeed, the eigenvalues of $K_i$ are $p(C_i|x)$ with multiplicity one and 0 with multiplicity $m_i - 1$. So, $\text{VNE}(K_i) = -p(C_i|x) \log p(C_i|x)$ (recall that for calculating VN entropy, we assume $0 \log 0 = 0$), and Properties 1 and 2 are fulfilled. $K$ is also symmetric and has non-negative eigenvalues. Thus, Property 3 is fulfilled as well.

Because $K$ satisfies all properties, we have proven that $\text{VNE}(K(s, x)) = \text{SE}(s|x)$.    □

**Theorem B.2** (KLE-c is more general than SE). *For any semantic clustering, there exists a kernel over semantic clusters $K_s(c, c')$ such that the VNE of this kernel is equal to semantic entropy (computed as in Eq. (5)).*

*Proof.* Analogously to Thm. B.1 but with the blocks of size one.    □

The theorems not only show that KLE generalizes SE but also provide an explicit form for a semantic kernel that can be used with KLE to recover SE.

# C    Kernel Hyperparameters

Following the discussion about kernel hyperparameters selection from Sec. 3.2, we visualize entropy convergence plots for Heat kernels in Fig. 2 and visualize heat and Matérn kernels on 2-d grid in Fig. C.4. Next, we expand on the question of parameter sensitivity, in Fig. C.3, and whether it is necessary to use a validation set for selecting kernel hyperparameters. We observe that both with reasonable default choices ($t = 0.3$, $\alpha = 0.5$, $\nu = 1$, and $\kappa = 1$) and by selecting hyperparameters on a separate set of answers, we outperform the existing methods. When choosing hyperparameters, we also have included a boolean flag whether the graph Laplacian should be normalized, as, generally

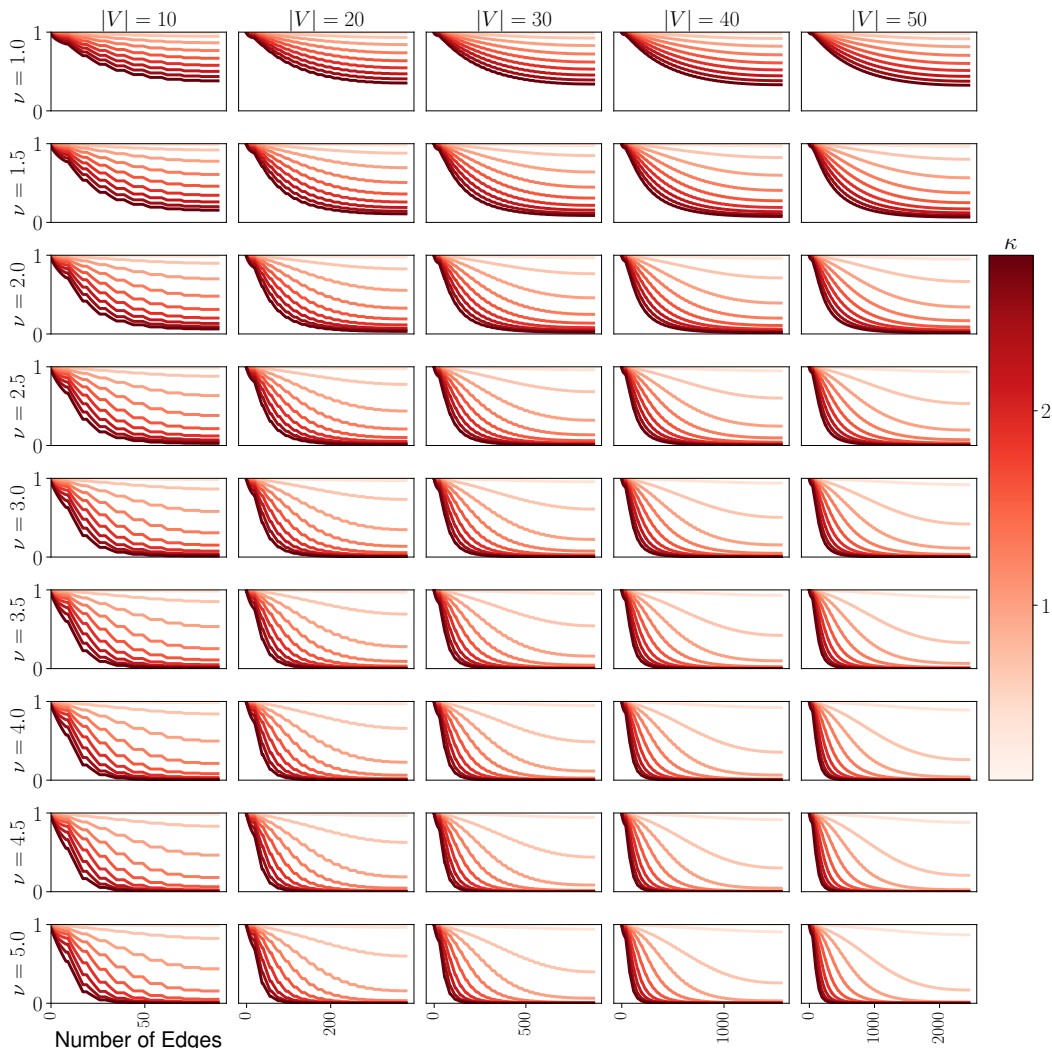

Figure C.1: Matérn Entropy Convergence Plots.

speaking, both the normalized and the standard graph Laplacians can be used with heat and Matérn kernels

$$L_{\mathrm{n}} = \left(D^+\right)^{1/2} L \left(D^+\right)^{1/2}, \tag{C.1}$$

where $D^+$ is the Moore-Penrose inverse of the degree matrix $D$. We observe similar results when analyzing other semantic kernels.

**Prompts.** We prompt the models to generate full sentences as answers with the following prompt:
`Answer the following question in a single brief but complete sentence.`.

Also, we have used the following prompt to check the accuracy of the responses:

`We are assessing the quality of answers to the following question:`
`{question} \n The expected answer is: {correct_answer}. \n The proposed`
`answer is: {predicted_answer} \n Within the context of the question, does`
`the proposed answer mean the same as the expected answer? \n Respond only`
`with yes or no.\n Response:`

Here we mark placeholders with the orange color. Or, if several correct answers were provided, we have used the following prompt:

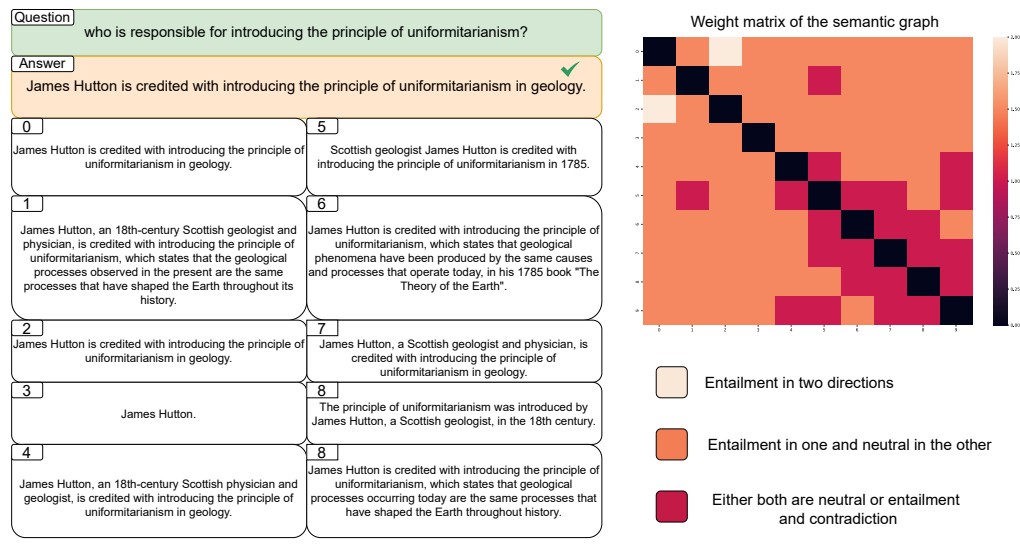

Figure C.2: Example from NQ.

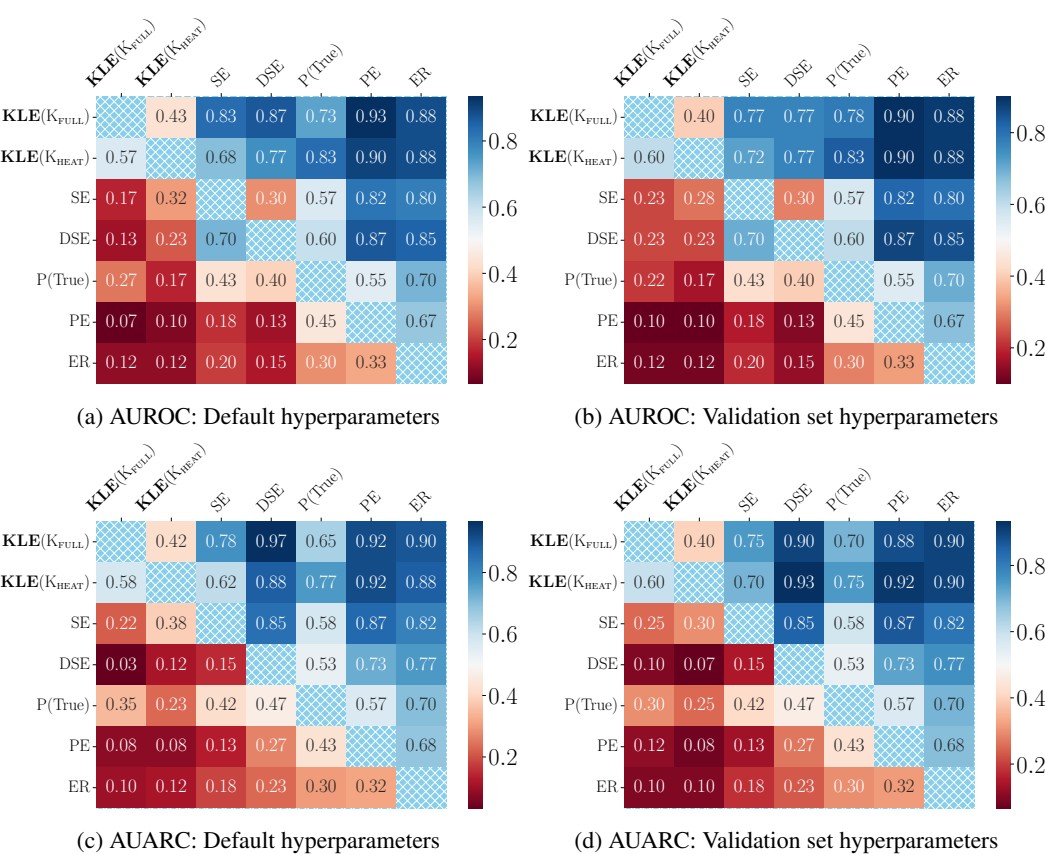

(a) AUROC: Default hyperparameters

(b) AUROC: Validation set hyperparameters

(c) AUARC: Default hyperparameters

(d) AUARC: Validation set hyperparameters

Figure C.3: Summary of **60** experimental scenarios. Comparing hyperparameters selection strategies. Our methods are labeled KLE($\cdot$).

```
We are assessing the quality of answers to the following question:
{question} \n The following are expected answers to this question:
{correct_answers}. \n The proposed answer is: {predicted_answer} \n Within
```

the context of the question, does the proposed answer mean the same as any of the expected answers? \n Respond only with yes or no.\n Response:

**Example.** We visualize an example from the NQ dataset in Fig. C.2; we have used Llama-2 70B Chat for this example. In order to analyze cases where SE and KLE are inconsistent, we ranked all the answers separately by KLE and SE and found those cases where the difference between indices in the list ranked by KLE and ranked by SE is high. In Fig. C.2, a model provides the correct answer. However, SE estimates the uncertainty to be high because it can detect only two answers as equal and thus considers the majority of the answers as semantically distinct. Instead, our method considers more fine-grained relations between the answers and provides bet-

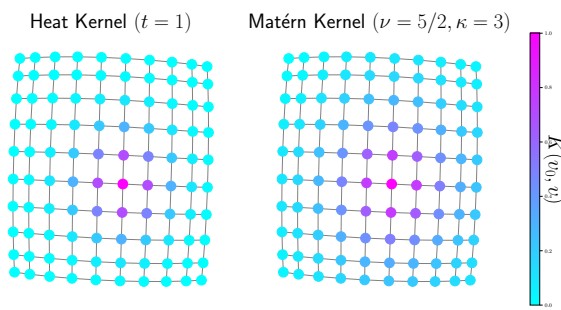

Figure C.4: Heat and Matérn kernels visualized on 2-d grid.

ter uncertainty estimates (i.e., orange and red cells in the weight matrix). It is an illustrative example of the cases we analyzed. It indicates that the longer and more nuanced the answers are, the more KLE would outperform SE.

## D Additional Experimental Details

In this section, we provide additional experimental results.

**Hardware and Resources.** We ran Llama 2 70B models on two NVIDIA A100 80GB GPUs, and the rest of the models on a single NVIDIA A100 80GB. The generation process took from one to seven hours (depending on a model) for each experimental scenario, and the evaluation additionally took roughly four hours per scenario which can be further optimized by reducing the number of hyperparameters. The project spent more resources due to other experiments. Our experimental pipeline first generates the answers for all the datasets and then computes various uncertainty measures. We did not recompute generations, but in each experimental run we only evaluated uncertainty measures.

**Licenses.** We release our code under a clear BSD-3-Clause-Clear. The datasets used in this paper are released under CC BY 2.5 (BioASQ; [35]), Apache 2.0 (TriviaQA; [29]), CC BY-SA 4.0 (SQuAD; [62]), MIT (SVAMP; [60]), and CC BY-SA 3.0 (NQ; [38]).

### D.1 Models and datasets

In Fig. D.1, we show samples from each dataset we used in the experimental evaluation of our method.

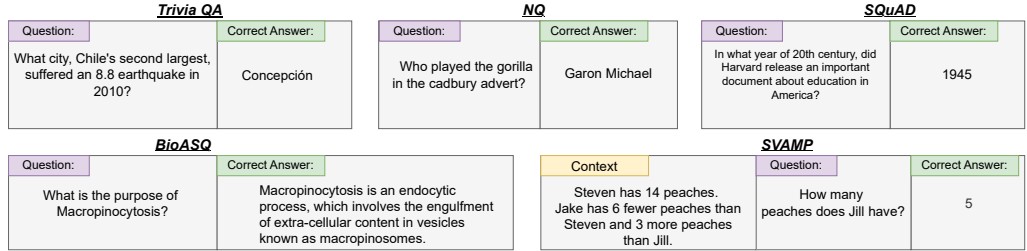

Figure D.1: Samples from datasets we use: Trivia QA, NQ, SQuAD, BioASQ, and SVAMP.

Additionally, we demonstrate the accuracy of the models used in the experiments on each dataset in Fig. D.2. As can be seen, we evaluate our method on a diverse set of models with a varying level of

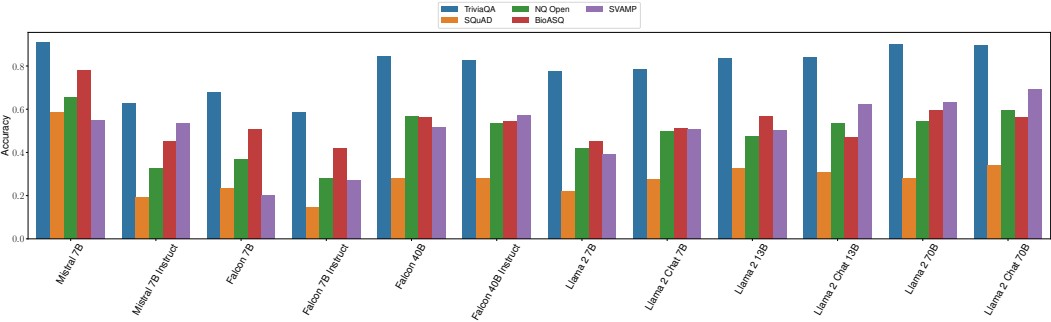

Figure D.2: Accuracy of the models

accuracy across the tasks at hand. This is especially important for UQ, because UQ methods should perform well for all the models regardless of their downstream effectiveness.

Real-world applications often involve deploying models with varying degrees of performance, and a robust UQ method should provide reliable uncertainty estimates for all of them. By demonstrating the efficacy of our method across a wide variety of models, we validate its applicability in diverse scenarios. This highlights that our approach can be confidently used in practical settings where model performance can fluctuate.

## D.2 Instruction-tuned and non-instruction-tuned models

Furthermore, we investigate the performance of UQ methods by splitting the set of experimental scenarios into instruction-tuned and non-instruction-tuned models. We visualize the splits in Fig. D.5. Interestingly, our approach significantly outperforms the existing methods when evaluated with instruction-tuned models, and only marginally outperforms when evaluated on non-instruction-tuned models. We can hypothesize that non-instruction-tuned models are better calibrated, and thus methods based on token-likelihoods perform well whereas instruction-tuning worsens calibration. This hypothesis is also supported by comparison of SE and DSE (DSE significantly outperforms SE on an instruction-tuned split, when AUROC is measured).

## D.3 Detailed results of UQ

We provide a detailed comparison of our method with previous uncertainty quantification measures. In Fig. D.3 and Fig. D.4, we show the results for a wide range of models across five datasets for non-instruction-tuned and instruction-tuned models, respectively. We want to note that ER has failed for Llama 2 13B (non-instruction-tuned version) for all datasets except BioASQ because training datasets for ER contained samples of only one class. We have assigned zero scores to the failed cases.

## D.4 NLI models accuracy

In Supplementary Note 2, Farquhar et al. [16] analyze the accuracy of various NLI models. They report that DeBERTa shows an average agreement of 0.8 with human raters, compared to an agreement of 0.87 between human annotators. We hypothesize that using a more advanced but computationally expensive NLI model, such as GPT-4, could improve the semantic kernel and, consequently, enhance uncertainty estimation using KLE.

# E Additional Notes

## E.1 Lexical, semantic, and syntactic variability

We resort to the 6-level model of the structure for text analysis proposed in [14] to extensively describe aspects of language beyond semantics. This model distinguishes four basic notions for text analysis: medium of transmission, grammar, semantics, and pragmatics. Medium of transmission is irrelevant to the study of language model outputs (however, it becomes relevant for multimodal

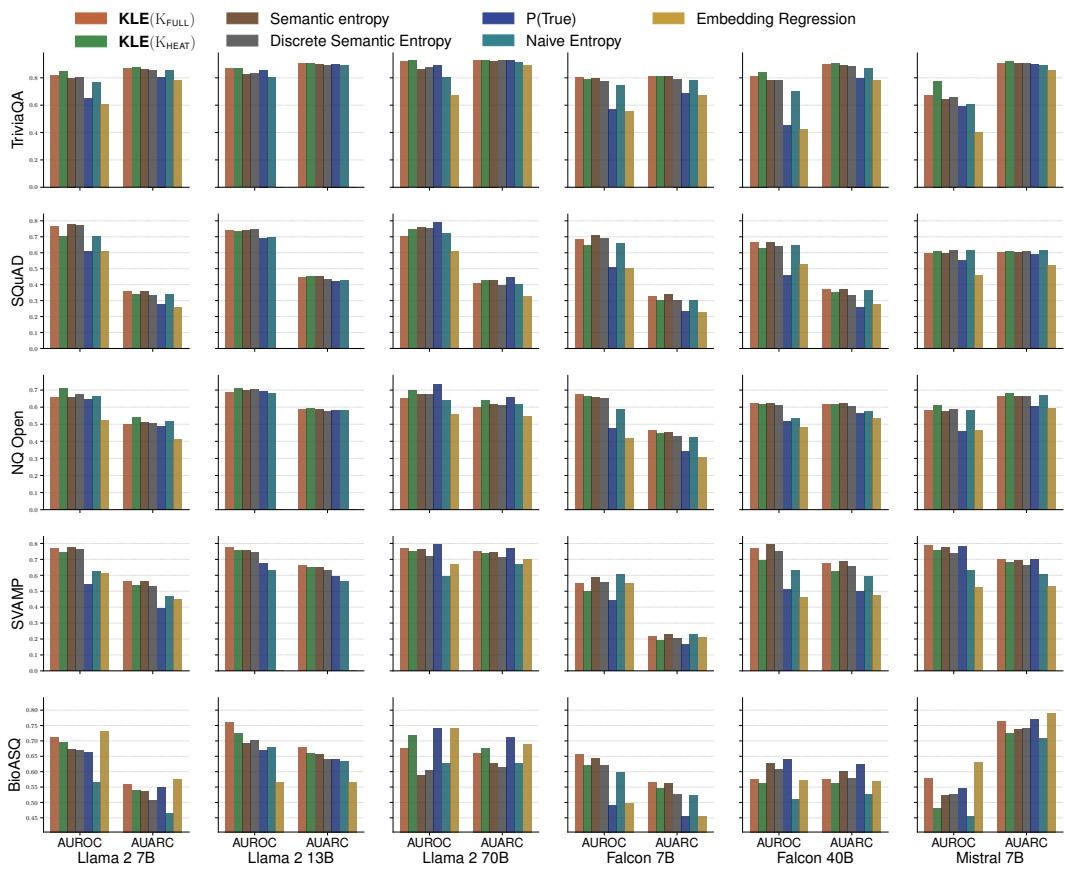

Figure D.3: Full results of non-instruction-tuned models

Table 2: Examples of semantic, syntactic, and lexical variability of a sentence "Paris is the capital of France."

| | Semantic Variability | Syntactic Variability | Lexical Variability |
|---|---|---|---|
| Paris is the capital of France. | Rome is capital of France
Paris is the capital of Italy. | The capital of France is Paris. | France's capital is situated in Paris.
France's capital city is Paris. |

foundation models that can, for instance, answer a request either with a text or an image); grammar is further divided into the syntax and morphology of the text and semantics into semantics and discourse. Another dimension is pragmatics, or how the text is used. In this work, we focus only on the semantics of the text. However, the method can be extended to other aspects of text analysis. For instance, one can design syntactic or pragmatic kernels. We leave the study of other kernel modalities to future works.

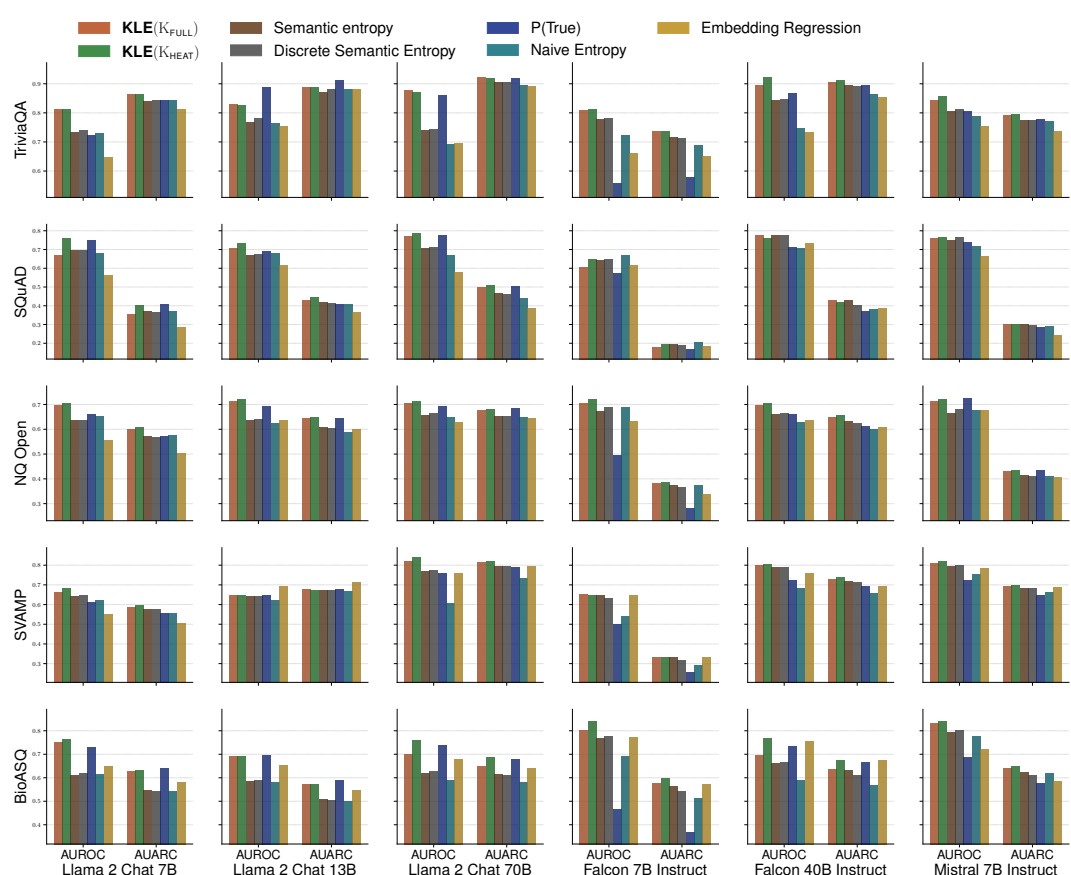

Figure D.4: Full results of instruction-tuned models

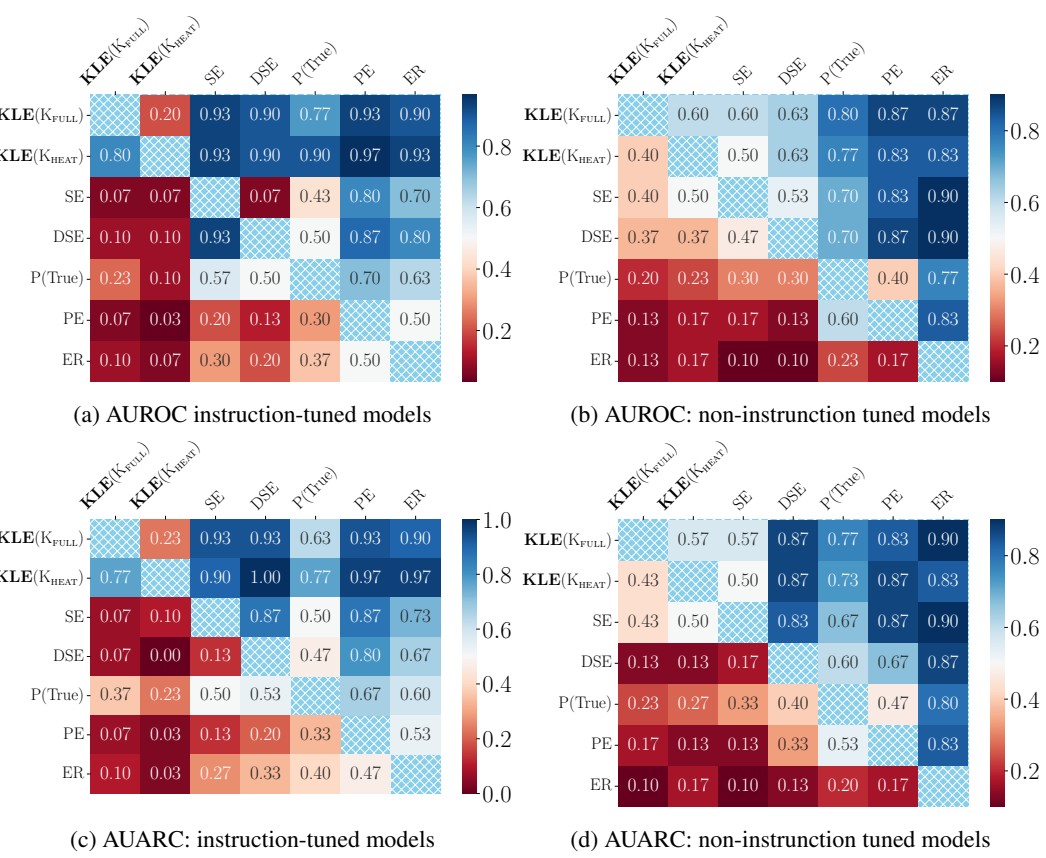

Figure D.5: Summary of **60** experimental scenarios. Comparing the results on instruction-tuned and non-instruction-tuned models. Our methods are labeled KLE(·).

