# OpenReview forum: "Kernel Language Entropy: Fine-grained Uncertainty Quantification for LLMs from Semantic Similarities"
_NeurIPS.cc/2024/Conference — NeurIPS 2024 poster_

### Official Review · Reviewer_tEe6 · 2024-07-09

**Soundness:** 3
**Presentation:** 4
**Contribution:** 3
**Rating:** 7
**Confidence:** 5

**Summary:**

The authors propose a novel Kernel Language Entropy (KLE) method for uncertainty estimation in white- and black-box LLMs. KLE defines positive semidefinite unit trace kernels to encode the semantic similarities of LLM outputs and quantifies uncertainty using the von Neumann entropy. It considers pairwise semantic dependencies between answers (or semantic clusters), providing more fine-grained uncertainty estimates than previous methods based on hard clustering of answers. We theoretically prove that KLE generalizes the previous state-of-the-art method called semantic entropy and empirically demonstrate that it improves uncertainty quantification performance across multiple natural language generation datasets and LLM architectures.

**Strengths:**

1. The authors propose Kernel Language Entropy, a novel method for uncertainty quantification in natural language generation.

2. The authors propose concrete design choices for our method that are effective in practice, for instance, graph kernels and weight functions.

3. The authors empirically compare our approach against baselines methods across several tasks an LLMs with up to 70B parameters (60 scenarios total), achieving SoTA results.

**Weaknesses:**

1.  The motivation is clear in Fig 1, but why choose the kernel to solve the problem, this part is not very clear.

2. The proposed method looks like it depends on the sampling times, which is expensive. How is the performance when we sample few times?

**Questions:**

1. How do I get the final answer based on 10 samplings? For example, in fig 1, what could be the final answer of LLM2, majority vote?

**Limitations:**

the authors adequately addressed the limitations

---

> ### Author Rebuttal · Authors · 2024-08-07
>
> Dear Reviewer tEe6,
>
> Thank you for the positive assessment of the novelty, practical effectiveness, and empirical comparison of our work. We would like to address the concerns and questions you raised below:
>
> **Weaknesses**
>
> >The motivation is clear in Fig 1, but why choose the kernel to solve the problem, this part is not very clear.
>
> Considering only semantic clusters, as done in semantic entropy [1], is severely limiting because it assigns responses to strictly separate equivalence clusters. In reality, the space of semantic meanings is more nuanced and fine-grained; some answers may be similar even though they are not exactly semantically equivalent. Therefore, introducing a metric to capture semantic similarity is essential, and assigning a kernel is a natural choice for this purpose. In the center of Figure 1, we visualize semantic kernels – those kernels characterize the distance in semantic space and thus are more expressive than simply considering a distribution over distinct semantic clusters. The von Neumann Entropy over the normalized kernel provides a convenient way to quantify entropy while accounting for the semantic similarity. We will clarify this in the next revision
>
> > The proposed method looks like it depends on the sampling times, which is expensive. How is the performance when we sample few times?
>
> For the rebuttal, we conducted an ablation study to investigate the optimal number of samples. See Fig.1 in the attached PDF. We observed that the performance of KLE is better than that of SE for all sample sizes, with a significant increase when the number of samples goes from 2 to 6, but continuing to grow until 10 samples. In practice, we recommend selecting as many samples as feasible and parallelizing sampling if needed.
>
> **Questions**
>
> > How do I get the final answer based on 10 samplings? For example, in fig 1, what could be the final answer of LLM2, majority vote?
>
> Following previous works [1, 2], we chose an answer sampled with a low temperature (T=0.1) as a final answer of an LLM, LL: 282-283. The majority vote is another viable option for short generations, however, in long-form generation answers become more diverse, which means that each semantic cluster is small and the majority vote becomes unreliable.
>
> **Concluding remarks**
> We would be grateful if you could let us know whether our explanations have addressed your concerns. Please let us know if you have any other questions or concerns.
>
> **References**
> [1] Farquhar, et al. (2024). Detecting Hallucinations in Large Language Models Using Semantic Entropy.
> [2] Kuhn, et al. (2023). Semantic Uncertainty: Linguistic Invariances for Uncertainty Estimation in Natural Language Generation.

---

> > ### Comment · Reviewer_tEe6 · 2024-08-11
> >
> > Thanks for the detailed response. All of my concerns are addressed. And after reading other reviews, I will keep my score.

---

### Official Review · Reviewer_DkpB · 2024-07-11

**Soundness:** 3
**Presentation:** 2
**Contribution:** 3
**Rating:** 7
**Confidence:** 4

**Summary:**

The paper introduces the method "Kernel Language Entropy", capturing semantic similarities of output sequences via semantic kernels and subsequently estimating uncertainty using the von Neumann entropy.

**Strengths:**

- The paper proposes a novel approach to estimate uncertainty in LLMs.
- It presents a solid theoretical foundation by showing that Kernel Language Entropy generalizes Semantic Entropy.

**Weaknesses:**

The primary area for improvement is the paper's structure. The paper is quite difficult to follow due to definitions and expressions not being clearly contextualized, for instance:
- **Sections and Headings**: Section 3 is inconsistent in its use of subsections and subheadings. It begins with a motivating example (subheading), followed by the formal definition of PSD kernels without a subheading. The definition of semantic kernels is embedded in the text without a subheading, making it less prominent than the practical approach for constructing semantic kernels, which has a dedicated subsection at the end of the section. Also, VNE is given more prominence than the main concept, KLE (starting with the definition of KLE (subheading) as the VNE, followed by deriving its properties, would improve coherence).
- **Kernels**: In Section 3.1, two explicit kernels ($K_{heat}$ and $K_{Matérn}$) are introduced. In Section 5, the authors then propose to use $K_{heat}$ and $K_{full}$, with $K_{full}$ being a weighted sum of $K_{heat}$ and $K_{SE}$. The previously proposed $K_{Matérn}$ is no longer considered in the main experiments, while $K_{SE}$ has not been introduced before. It only becomes clear in Section B of the Appendix that $K_{SE}$ is equal to Semantic Entropy, which is referred to as $SE$ in Section 5. Also, the importance of the weighting factor for $K_{full}$ is not discussed. In general, the criteria for choosing between kernels are unclear (and not all kernels consistently outperform the baselines across datasets).

**Questions:**

- Is the rather small model Llama 3 8B Instruct capable of evaluating the correctness of an output sequence? Have you considered other (statistics-based) metrics to evaluate the correctness?
- Have the authors considered utilizing a regression model that directly assigns a single value to the semantic similarity instead of unintuitively having to aggregate the three classes of the NLI model?

**Limitations:**

The authors adequately addressed limitations.

---

> ### Author Rebuttal · Authors · 2024-08-07
>
> Dear Reviewer DkpB,
>
> Thank you for your positive assessment of novelty and theoretical motivation of our work. Please, let us address your questions and pointed weaknesses:
>
> **Weaknesses**
> >Section 3 is inconsistent in its use of subsections and subheadings. It begins with a motivating example (subheading), followed by the formal definition of PSD kernels without a subheading […]
>
> Thank you for pointing this out! We will add subheading to the KLE definition and further add a subsection to highlight “Semantic Kernels and KLE” in order to improve the structure of Section 3.
>
> > In Section 3.1, two explicit kernels ($K_{Heat}$ and $K_{Matern}$) are introduced. In Section 5, the authors then propose to use $K_{Heat}$ and $K_{Full}$ , with being a weighted sum of $K_{Heat}$ and $K_{SE}$. The previously proposed $K_{Matern}$ is no longer considered in the main experiments, while has not been introduced before. It only becomes clear in Section B of the Appendix that is equal to Semantic Entropy, which is referred to as in Section 5. Also, the importance of the weighting factor for is not discussed. In general, the criteria for choosing between kernels are unclear (and not all kernels consistently outperform the baselines across datasets).
>
> Thank you for bringing it to our attention! $K_{\operatorname{Matern}}$ was considered in the main experiments and showed similar results to $K_{\operatorname{heat}}$ (see Fig. 4). Thank you for pointing out the inconsistencies regarding  $K_{\operatorname{SE}}$; we will include its definition in the main text. We chose a weight factor using the validation set, and since the results with the validation set versus the default parameters are not significantly different, it is reasonable to use the default value (0.5) in practice or choose it as a hyperparameter with a validation set. It appears that all kernels outperform SE, with the best results observed using the vanilla $K_{\operatorname{heat}}$ (see Fig. 4). $K_{\operatorname{heat}}$ consistently shows strong performance and can be chosen as a default choice. We will add more details on this to avoid any confusion.
>
> **Questions**
> >Is the rather small model Llama 3 8B Instruct capable of evaluating the correctness of an output sequence? Have you considered other (statistics-based) metrics to evaluate the correctness?
>
> We evaluated the Llama-2-70B-chat results using GPT-4 on TriviaQA and found the accuracy assessment to be consistent with Llama-3-8B in 95% of cases. Additionally, we compared Llama-3-8B with human annotations of accuracy and found agreement in 90% of cases. The source code for the experiments is available, allowing users to re-run them with GPT-4 if their budget permits. However, we chose to use an open-source model to support researchers with limited budgets. Overall, Llama-3-8B appears well-suited for this task, accessible, and easy to use even with limited GPU access. We will expand on this in the next revision!
>
> > Have the authors considered utilizing a regression model that directly assigns a single value to the semantic similarity instead of unintuitively having to aggregate the three classes of the NLI model?
>
> We considered several ideas on how semantic kernels can be enhanced, including utilizing a regression model. However, in this work, we chose to focus on building upon existing approaches, and previously, NLI model outputs were used directly. We also employed confidences from the NLI model as graph weights but did not observe the improvement ($K^{DB}_{*}$, Fig. 4). We leave the improvement of the similarity measures for future work and briefly discuss these potential improvements and their limitations in LL: 354-356. We will expand on this in the next revision!
>
> **Concluding remarks.**
> We would be grateful if you could let us know whether our explanations have addressed your concerns. Please let us know if you have any other questions or concerns.

---

> > ### Comment · Reviewer_DkpB · 2024-08-08
> >
> > Thank you for the rebuttal. Addressing the mentioned points in the final version of the paper will indeed improve its clarity and overall quality. I have no further questions and am updating my score.

---

### Official Review · Reviewer_Raee · 2024-07-12

**Soundness:** 4
**Presentation:** 3
**Contribution:** 3
**Rating:** 6
**Confidence:** 2

**Summary:**

In this work, the authors propose Kernel Language Entropy, which shares a similar concept to semantic uncertainty but additionally considers semantic similarity. Based on this proposed theory, they design graph kernels and weight functions to estimate LLM uncertainty. Furthermore, they demonstrate that their method generalizes semantic entropy. The empirical results confirm the effectiveness of the proposed method across various tasks and LLMs.

**Strengths:**

Originality: This paper extends semantic entropy to Kernel Language Entropy, which additionally takes into account the semantic similarity between clusters. The idea is well-motivated.

Quality: This paper provides detailed theoretical analysis and designs several variants. The experimental results demonstrate the effectiveness of these designs. The experiments are extensive.

Clarity: This paper is well-structured.

Significance: This paper focuses on estimating LLM uncertainty, which is significant for LLM applications, as they often make mistakes in their responses. The techniques proposed in this work could be beneficial in addressing this issue.

**Weaknesses:**

1. KLE requires iterative sampling from the LLM, which is computationally costly. This leads to delayed responses and limits its applicability.

2. It seems that this method can only estimate the confidence regarding whether the LM will correctly answer the query, but it cannot predict the confidence for a given answer. For example, in scenarios like ranking candidate answers, we can use P(True) or PE to estimate answer confidence, which KLE cannot do.

**Questions:**

1. Which sample do you choose as the final answer among multiple samples, the answer in the biggest cluster or just a random one? I did not find this detail (perhaps I missed it), but it is important.

2. Since the answer correctness is determined by an 8b model, I am curious about how reliable the predicted label is. From my previous experience, even GPT-3.5 is not capable of reliably estimating the correctness of model responses, especially when the responses are phrases. Tian et al. [1] also noted a similar issue, as seen in their Appendix C.

[1]. Just Ask for Calibration: Strategies for Eliciting Calibrated Confidence Scores from Language Models Fine-Tuned with Human Feedback.

3. Can this method be applied to long-form responses? Considering that long-form responses typically consist of multiple claims.

4. Missing the following related work:

Language Models (Mostly) Know What They Know

LitCab: Lightweight Language Model Calibration over Short- and Long-form Responses

**Limitations:**

The authors acknowledge the issue of computational cost. Investigating ways to elicit confidence from the LLM itself could be a possible direction for addressing this.

---

> ### Author Rebuttal · Authors · 2024-08-07
>
> Dear reviewer Raee,
>
> Thank you for your positive evaluation of our work and for highlighting its originality, quality, clarity, and significance. We hope to address your concerns and questions in our response below.
>
>
> **Weaknesses**
> >KLE requires iterative sampling from the LLM, which is computationally costly. This leads to delayed responses and limits its applicability.
>
>
> Our method, like other most effective methods such as semantic entropy (SE) and discrete semantic entropy (DSE), also requires sampling multiple answers. One way to overcome this is by generating responses in parallel, which helps to avoid the delay but uses more resources. In Fig. 1 in the rebuttal PDF, we include an ablation study on the number of samples for NQ and BioASQ. We briefly discuss this limitation in LL: 351-352.
>
> >It seems that this method can only estimate the confidence regarding whether the LM will correctly answer the query, but it cannot predict the confidence for a given answer. For example, in scenarios like ranking candidate answers, we can use P(True) or PE to estimate answer confidence, which KLE cannot do.
>
> Yes, that is correct! Methods like SE and KLE estimate the predictive semantic entropy of a model, which is different from estimating the confidence. Estimating uncertainty as predictive entropy is also an important problem for many applications (e.g., classification with rejection), and was used extensively in Bayesian deep learning. We will discuss these considerations in the limitations section.
>
> >Which sample do you choose as the final answer among multiple samples, the answer in the biggest cluster or just a random one? I did not find this detail (perhaps I missed it), but it is important.
>
> Following prior literature [1], we select a low temperature sample (T=0.1) in our experiments to quantify accuracy (LL: 282-283).
> > Since the answer correctness is determined by an 8b model, I am curious about how reliable the predicted label is. From my previous experience, even GPT-3.5 is not capable of reliably estimating the correctness of model responses, especially when the responses are phrases. Tian et al. [1] also noted a similar issue, as seen in their Appendix C.
>
> We evaluated the Llama-2-70B-chat results using GPT-4 on TriviaQA and found the accuracy assessment to be consistent with Llama-3-8B in 95% of cases. Additionally, we compared Llama-3-8B with human annotations of accuracy and found agreement in 90% of cases. The source code for the experiments is available, allowing users to re-run them with GPT-4 if their budget permits. However, we used an open-source model to support researchers with limited budgets. Overall, Llama-3-8B appears well-suited for this task, accessible, and easy to use even with limited GPU access.
> >Can this method be applied to long-form responses? Considering that long-form responses typically consist of multiple claims.
>
> Yes! As a matter of fact, our method particularly excels when working with longer answers. In our experimental setup, as illustrated by the examples in Appendix (Fig. C.2), the generated responses include the main answer as well as additional contextual content. For example, instead of simply answering “Laplace,” our LLMs might respond with “Laplace, a French scientist who studied…” By building a semantic kernel, we capture these fine-grained similarities and estimate uncertainty more effectively. In contrast, SE aims to assess the equivalence between answers, and including additional information often leads to worse performance (each answer tends to have its cluster). This is demonstrated in Fig. C.2 and discussed in the last paragraph of Appendix C.
> > Missing the following related work …
>
> Thank you! We will add these references.
>
>
> **Concluding remarks**
> We would be grateful if you could let us know whether our explanations have addressed your concerns. Please let us know if you have any other questions or concerns.
>
> **References**
> [1] Kuhn, et al. (2023). Semantic Uncertainty: Linguistic Invariances for Uncertainty Estimation in Natural Language Generation.

---

> > ### Comment · Reviewer_Raee · 2024-08-12
> >
> > Thanks for the detailed response. I will keep my positive score.

---

### Official Review · Reviewer_S5q9 · 2024-07-15

**Soundness:** 3
**Presentation:** 2
**Contribution:** 2
**Rating:** 4
**Confidence:** 4

**Summary:**

This paper is highly motivated by and heavily draws from Kuhn et al. (ICLR, 2023) “Semantic Uncertainty: Linguistic Invariances for Uncertainty Estimation in Natural Language Generation.” In Kuhn’s original paper they propose an unsupervised way to calculate the semantic uncertainty of LLMs by (1) generating a set of outputs from an LLM, (2) clustering these outputs by semantic equivalence via an NLI model, and (3) plugging in these clusters to an equation for “semantic entropy.”

This paper proposes an extension of the strict clusters of Kuhn et al.~by creating a “semantic graph” and calculate the “graph kernel” for the uncertainty metric. Thus, the authors attempt to measure the semantic entropy *between* clusters rather than just *within* clusters (the latter which is the approach of the previous work, Kuhn et al. (2023)).

The authors provide a lot of math and theory trying to justify that their generalization from Kuhn et al. is meaningful. However, the empirical experiments show modest to no performance gains from their generalization on real-world QA datasets and LLMs.

**Strengths:**

1. The authors have crafted a very extensive empirical set-up with 12 different LLMs on five datasets. They also get 10 LLM outputs per input and also obtain confidence intervals over 1000 bootstrap resamples (line 286-287). They combine this with five different baseline models for comparison to their proposed methods. I commend the authors on this extensive and painstaking experimental set-up.

2. The authors seem very well-versed in the previous literature and identify a clear gap in Kuhn et al.: that the “hard clustering” of Kuhn et al. misses what could be a softer clustering of semantic similar (which the authors tackle by constructing a semantic graph with weights between clusters).

**Weaknesses:**

1. **The claims are not backed by strong enough evidence.**

On line 59, the authors claim their approach achieves SoTA results. However, the empirical results (Table 1) are little to modest gains over baselines. Additionally, the bolding in Table 1 is misleading. Results should only be bolded if the method’s confidence interval is non-overlapping the confidence intervals of other methods (which they are not). For example, in Table 1 for AUROC on BioASQ, $0.88 \pm 0.03$ for KLE (K_FULL) is overlapping with P(True) at $0.86 \pm 0.03$. Likewise for SVAMP AUROC with $0.77 \pm 0.02$ for KLE (K Heat) and ER at $0.75 \pm 0.02$. This also makes me skeptical of how the authors are calculating the “win rates” for the other figures as well.

2. Additionally the authors do not investigate the **error/accuracy of intermediate models** that the experimental set-up relies upon.

A key part of the authors pipeline is using DeBERTa-Large-MNLI to predict whether LLM outputs entail one another and create their semantic graph (line 262). However, DeBERTa-Large-MNLI is also an imperfect model. What was the accuracy of this model on the domains/datasets in the authors’ empirical pipeline? Are there ways to propagate uncertainty from this intermediate NLI model downstream to the final uncertainty calculations of other LLM’s outputs?

3. Additionally, see the questions below.

**Questions:**

1. In the abstract, you motivate this work by saying “by detecting factually incorrect model responses, commonly called hallucinations.” Yet, you never actually show that you detect hallucinations (which require world knowledge that is external and more complex than the “semantic uncertainty” you’re actually targeting). How do you justify this disconnect?

2. On line 25 you cite two works that you use to justify “As LLM predictions tend to be well-calibrated.” However there is an *enormous* body of literature that shows the opposite. I would recommend hedging on this statement or providing citations of the opposite (non-calibration) findings.

3. Figure 1 lacks some details and motivation. In this example, I would not expect the named entity outputs, (e.g., “Laplace” or “Kolmogorov and Laplace”) to have any other lexical variants that have similar semantic meaning because these are named entities. In this figure, what does the little boxes next to “Semantic Kernels” represent? I would recommend explaining this more (e.g., I think you’re implying that blue means more similarity between particular clusters?)

4. In Figure 3’s caption, please explain why only “values larger than or equal to 0.62” correspond to a p-value less than 0.05.

5. I would recommend explaining how you are calculating AUPRC and why it is important. I know Kuhn et al. also use this metric but you do not make clear what this metric is in your standalone work. From re-reading Kuhn et al., I took away that they calculated an “entropy score” to predict whether a model’s answer to a question is correct. They vary the threshold of the entropy score for correctness prediction to get AUROC.

**Limitations:**

Addressed.

---

> ### Author Rebuttal · Authors · 2024-08-07
>
> Dear reviewer S5q9,
>
> Thank you for a thoughtful and constructive review. We are pleased to hear that you found the experimental setting and the research problem in our work interesting. We hope to address your concerns in our reply below.
>
> **Weaknesses**
> > […] the empirical results (Table 1) are little to modest gains
>
> In Tab. 1, our method is the best or among the best for all datasets and both models, while the second-best method is different for different setups (ER, P(True), or SE). **Statistical significance is achieved primarily by repeating the experiment with many different models and datasets.** Tab. 1 shows results for only the two largest models out of 12. **See Fig. 3 for the summary of all experiments.** Statistical significance was calculated by the sign test [1] across the 60 experiments, and confirmed in Fig. 3 that the differences between our method and any other method are always statistically significant (LL: 285-290). To further explicitly demonstrate the size of the improvement, we show the relative gains in AUROC and AUARC in Tab. 1 in the rebuttal PDF.
> > the bolding in Table 1 is misleading
>
> It is common to bold the results when the average is the best, e.g. https://arxiv.org/pdf/2106.10934 or https://arxiv.org/pdf/2011.13456, but we are happy to add a comment to avoid confusion.
> >[...] how the authors are calculating the “win rates” [...]
>
> Win-rates represent the fraction of cases where one method outperforms another, based on a better mean value of the corresponding metric (LL: 285-290).
> > authors do not investigate the error/accuracy of intermediate models [...]
>
> - Models for generating answers: We report their accuracy in Appendix D (Fig. D2).
> - NLI models: The performance of NLI models in the same setting is analyzed by [2]. We will add a reference and summarize their analysis.
> - Models for checking the answers: We compared the performance of Llama-3-8B and GPT-4 for TriviaQA by Llama-2-70B-chat and observed 95% agreement. We additionally measured the performance compared to humans and observed 90% agreement in 100 cases.
>
> > DeBERTa-Large-MNLI analysis and uncertainty propagation
>
> The uncertainty from the NLI model can be propagated to the final KLE model. In our experiments, we have used kernels where NLI confidences directly form weights (shown in Fig. 4 under $K^{DB}_{*}$). Moreover, KLE can combine multiple NLI models via kernel composition. We leave this and other alternatives to constructing better semantic kernels for future work (LL: 354-356). The accuracy of NLI in this setting is assessed by [2], which we will comment on.
>
> **Questions**
>
> > you motivate this work by saying “by detecting factually incorrect model responses, commonly called hallucinations.” Yet, you never actually show that you detect hallucinations [...]
>
> We use predicted uncertainty to detect factually incorrect responses in our experiments, following the evaluation method used in [3] and other studies. AUROC and AUARC measure the effectiveness of uncertainty in detecting hallucinations (see LL: 273-280 and [2]).
> > you cite two works that you use to justify “As LLM predictions tend to be well-calibrated.” [...] I would recommend hedging on this statement or providing citations of the opposite
>
> Thanks! We will discuss these papers. We will further clarify the differences in observations regarding calibration of LLMs (e.g., base vs. instruction-tuned models).
> Importantly, note that KLE does not rely on calibration and performs well when an LLM is not well-calibrated (see Fig. D5).
> >Fig. 1 […] I would not expect the named entity outputs [...] to have any other lexical variants [...] what does the little boxes next to “Semantic Kernels” represent? [...]
>
> Thank you for pointing this out! The 3x3 matrices represent kernels over clusters, where colors show kernel values. Non-diagonal elements indicate similarities, and diagonal elements represent $p(C_i | x)$, similar to theoretical proofs. We'll add a more detailed explanation.
> The answer "Laplace" can appear in different forms, like "Pierre-Simon Laplace" or “Laplace, a French scholar.” Answers in each cluster may or may not vary lexically.
> Fig. 1 shows that even with identical distributions over semantic clusters, the kernels differ. The KLE method offers better uncertainty quantification than SE (right side).
> >In Fig. 3 [...] why only “values larger than or equal to 0.62” correspond to a p-value less than 0.05.
>
> A sign test shows that method A outperforms method B if the value is ≥ 0.62, and method B outperforms method A if the value is ≤ 0.38, according to the sign test. Values between 0.38 and 0.62 are not statistically significant. We will clarify this in the next revision.
> >explain how you are calculating AUPRC and why it is important. […] they calculated an “entropy score” to predict whether a model’s answer to a question is correct
>
> Similar to [2, 3], we use the entropy score to predict the correctness of the generated responses. The AUROC and AUARC measure the ability of uncertainty to detect hallucinations (LL: 273-280).
>
> **Concluding remarks**
>
> We appreciate your comments on our empirical results. We hope our answers have fully addressed your concerns.
>
> We would like to emphasize that our method is both empirically effective and theoretically sound. It holds methodological value and opens the potential for developing semantic kernels for other LLM outputs, such as structured data or mathematical proofs, thereby broadening the scope of uncertainty quantification.
>
> We would be grateful if you could let us know whether our explanations have addressed your concerns. Please let us know if you have any other questions or concerns.
>
> **References**
> [1] Dixon, et al. (1946). The Statistical Sign Test.
> [2] Farquhar, et al. (2024). Detecting Hallucinations in Large Language Models Using Semantic Entropy.
> [3] Kuhn, et al. (2023). Semantic Uncertainty: Linguistic Invariances for Uncertainty Estimation in Natural Language Generation.

---

> > ### Comment · Reviewer_S5q9 · 2024-08-07
> > **Response**
> >
> > > Statistical significance is achieved primarily by repeating the experiment with many different models and datasets
> >
> > "Statistical significance" has a precise, technical meaning as in [Berg-Kirkpatrick 2012](https://aclanthology.org/D12-1091.pdf).
> >
> > Per my comment "Results should only be bolded if the method’s confidence interval is non-overlapping the confidence intervals of other methods (which they are not)", in each of the 5 datasets, how many times did your method have better point estimates and non-overlapping CIs with the baselines' for *both* metrics?

---

> ### Author Response · Authors · 2024-08-12
>
> Dear reviewer S5q9, Thank you for your comment and for engaging in the discussion!
>
> >Per my comment "Results should only be bolded if the method’s confidence interval is non-overlapping the confidence intervals of other methods (which they are not)", in each of the 5 datasets, how many times did your method have better point estimates and non-overlapping CIs with the baselines' for both metrics?
>
> We observe that the standard error from bootstrap for each evaluation is consistently large across all method-dataset pairs, and more related to the experimental setup than differences between the methods. Precisely because the CIs are overlapping in individual cases, we repeated the experiment for 60 model-dataset pairs. Our experimental setup follows a recent paper (https://www.nature.com/articles/s41586-024-07421-0, [2]), where the authors specifically emphasize:
> “We report the raw average score across held-out evaluation datasets without standard error because the distributional characteristics are more a property of the models and datasets selected than the method. **Consistency of relative results across different datasets is a stronger indicator of variation in this case.**”
>
> Additionally to evaluating such consistency with the binomial test (Fig. 3 and 4, main text), we’ve included relative mean gain in the reported metrics per dataset in the rebuttal PDF (Tab. 1).
>
> Finally, we report a comparison in each of the 5 datasets separately, when _both metrics are considered simultaneously and the mean estimate from the bootstrap_ is used for comparison. The table shows the numbers of wins, ties (one metric is better, another is worse), and losses of our method compared to the two strongest baselines (the metrics were strongly correlated and therefore highly consistent with each other). Each cell contains (#wins / #ties / #losses).
>
> | 	| SQUAD | SVAMP | NQ | TriviaQA | BioASQ | Total wins |
> | -------- | ------- | ------- | ------- | ------- | ------- | ------- |
> | SE | 5/2/5 | 5/1/6 | 10/1/1 | 11/1/0 | 9/0/3 | 42.5 ( p ≤ 0.001) |
> | P(True) | 10/1/1 | 9/1/2 | 9/2/1 | 10/1/1 | 6/2/4 | 47.5 (p < 0.00001) |
>
> The p-value was calculated using the sign test by splitting the ties between wins and losses [1].
>
> In summary, _while the results for a single dataset may not always be conclusive, we in the end observe very strong evidence that our method performs overall the best by repeating the experiment over 60 experimental scenarios_.
>
> We hope that we have addressed your remaining concern!
>
>
> ## References
> See the references from our response above.

---

> > ### Comment · Reviewer_S5q9 · 2024-08-12
> > **Response**
> >
> > Thanks for the detailed response. I changed my score to a 4.

---

### Author Rebuttal · Authors · 2024-08-07

We thank all reviewers for their thoughtful reviews, valuable suggestions, and for taking the time to read our paper!

We particularly appreciate the positive recognition of many aspects of our work, including its novelty (tEe6, DkpB, Raee), significance (tEe6, Raee), empirical comparison and experimental setup (tEe6, S5q9), and theoretical results and research problem (DkpB, S5q9).
We hope we have addressed all questions and concerns raised by the reviewers and are happy to discuss any remaining concerns or questions during the rebuttal.

**Main Changes:**
- Improving the clarity of the text with minor fixes and additional explanations.
- Including additional results that explicitly show the relative improvement in AUROC and AUARC compared to other models across 60 experimental scenarios setups (see Tab. 1 in the attached PDF). (S5q9)
- Investigating the impact of the sample size with a new ablation study (see Fig. 1 in the attached PDF). (tEe6, DkpB)
- Confirming the validity of using Llama-3-8B for accuracy checking by comparing it with human evaluation and GPT-4. (Raee, tEe6, DkpB, S5q9)

Please, take a look at the attached PDF for visualizations and tables.

We would be grateful if you could let us know whether our explanations have satisfactorily addressed your concerns. We are also open to discussing any other questions you may have.

---

### Decision · Program_Chairs · 2024-09-25

**Decision:**

Accept (poster)

**Comment:**

This paper provides an improvement over Kuhn et al. (ICLR, 2023) "Semantic Uncertainty: Linguistic Invariances for Uncertainty Estimation in Natural Language Generation", which performs hard clustering of the generated answers and computes the entropy of the clusters, by modifying the notion of semantic entropy to consider inter-cluster interactions. This is done by considering instead the von Neumann entropy (VNE) with respect to a graph kernel, where the graph edge weights encodes the similarity between the clusters.

Most reviewers agree that this is a simple but well motivated and theoretically sound approach, pointing out as weaknesses the modest empirical improvements, the requirement of multiple samples to compute uncertainty, and the need to improve the structure of the paper. While this seems a relatively small contribution on top of Kuhn et al. and it has its limitations, the proposed approach seems solid and addresses an important problem with the original definition of semantic entropy by Kuhn et al. The rebuttal is convincing and provides additional results which will improve the paper. I am leaning to recommend acceptance.